# Understanding Transformer-based Vision Models through Inversion

## Abstract

Understanding the mechanisms underlying deep neural networks remains a fundamental challenge in machine learning and computer vision. One promising, yet only preliminarily explored approach, is feature inversion, which attempts to reconstruct images from intermediate representations using trained inverse neural networks. In this study, we revisit feature inversion, introducing a novel, modular variation that enables significantly more efficient application of the technique. We demonstrate how our method can be systematically applied to the large-scale transformer-based vision models, Detection Transformer and Vision Transformer, and how reconstructed images can be qualitatively interpreted in a meaningful way. We further quantitatively evaluate our method, thereby uncovering underlying mechanisms of representing image features that emerge in the two transformer architectures. Our analysis reveals key insights into how these models encode contextual shape and image details, how their layers correlate, and their robustness against color perturbations. These findings contribute to a deeper understanding of transformer-based vision models and their internal representations.

## 1 Introduction

In recent years, the focus in the field of computer vision has shifted from convolutional neural networks (CNNs) to transformer-based vision models (TVMs) (Dosovitskiy et al., 2020; Li et al., 2023b; Carion et al., 2020; Zhu et al., 2021; Zhang et al., 2022). Despite their impressive performance, the internal mechanisms that enable these networks to solve complex tasks remain largely opaque. This opaqueness prevents an interpretation and a clear understanding of how predictions are made within these networks (Zhang & Zhu, 2018; Fan et al., 2021; Li et al., 2022b). Enhancing network interpretability, i.e., understanding the mechanisms underlying the functionality of a particular deep neural network (DNN), is crucial for ensuring safety, optimizing performance, and identifying potential weaknesses.

Feature inversion using inverse networks, introduced by Dosovitskiy and Brox (Dosovitskiy & Brox, 2016), is an early technique to interpret the processing capabilities of DNNs for vision. Building on a substantial body of work on generating images from intermediate representations (Erhan et al., 2009; Zeiler & Fergus, 2014; Mahendran & Vedaldi, 2014; Springenberg et al., 2015), their method involved training an inverse network for each layer of the CNN AlexNet Krizhevsky et al. (2012) to reconstruct input images from intermediate representations. By analyzing these reconstructed images and their distinct characteristics from various layers, they gained insights into the underlying mechanisms of the architecture.

While feature inversion was successfully applied to AlexNet, it has not seen widespread adoption in the context of modern DNNs for vision. We attribute this limited use to two main factors. Firstly, training individual inverse networks for each layer of a DNN is computationally demanding, particularly for the large CNNs and TVMs of today. Secondly, the potential of using feature inversion as a tool for analyzing and interpreting neural networks was only preliminarily explored by Dosovitskiy and Brox (Dosovitskiy & Brox, 2016), leaving much of the of the capabilities of the method underutilized.

In this work, we revisit feature inversion and apply it to two widely used TVMs (see Figure 1 for an illustration): Detection Transformer (DETR) (Carion et al., 2020) and Vision Transformer (ViT) (Dosovitskiy et al.,

2020), which serve as standalone models as well as backbones in state-of-the-art vision systems (e.g., Li et al., 2022a; Oquab et al., 2024; Ravi et al., 2024). We begin by introducing a novel feature inversion approach that builds on the classical idea of training layer-wise inverse networks but adopts a modular strategy that inverts only specific components of the model. This design enables a substantially more efficient application of feature inversion to large-scale deep networks.

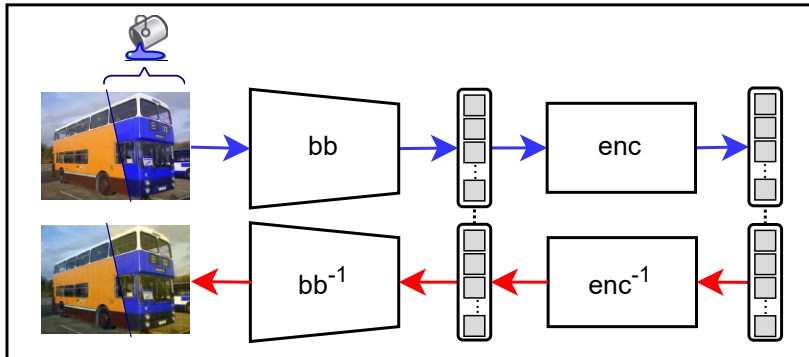 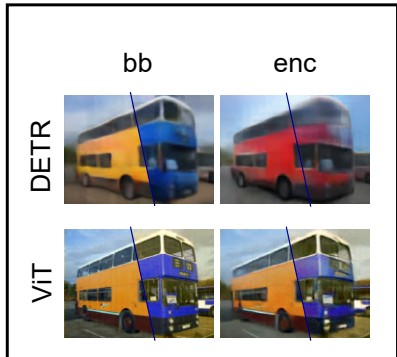

Figure 1: Illustration of our approach. **Left**: We invert components of transformer-based vision models, such as the backbone and encoder. **Right**: Using these inverted components, we reconstruct images from different processing stages to analyze DETR and ViT. Here, we recolor a yellow bus to blue to examine color processing in the two architectures.

We empirically validate our method on DETR and ViT, and demonstrate how reconstructed images from different layers can be interpreted in a systematic and meaningful way (see Figure 1 for an illustration). We introduce several novel analysis techniques for feature inversion and quantitatively evaluate a range of hypotheses about internal representations. Our analyses uncover key properties of DETR and ViT, as well as fundamental architectural differences. Among our findings are contrasts in the preservation of contextual shape and image detail, inter-layer representational correlations, and robustness to color perturbations. Notably, despite their similar architecture, the two models exhibit distinct strategies for visual abstraction: DETR gradually transforms object shapes and colors into more prototypical representations at higher layers, while ViT retains fine-grained visual detail throughout all layers. We summarize our core contributions as follows:

- We introduce a novel highly efficient feature inversion method based on modular, independently trained inverse components, which we empirically test and validate.

- We demonstrate how reconstructed images can be systematically used to interpret internal processing mechanisms, introducing new analysis techniques such as targeted embedding manipulation.

- We identify shared properties across DETR and ViT, including gradual representation changes across layers, thereby extending prior findings on ViT to the DETR architecture.

- We reveal fundamental differences between DETR and ViT, particularly in terms of image detail preservation, abstraction behavior, and robustness to color perturbations.

## 2 Related work

In computer vision, feature inversion is a technique that reconstructs an input image from its intermediate feature representations, enabling direct inspection of the information a DNN preserves, discards, or transforms at different processing stages. It serves as a general tool for studying information flow and representation dynamics of DNNs.

A seminal work by Mahendran & Vedaldi (2014) framed inversion as an optimization problem in the image space, generating images whose features matched those of a target layer in CNNs. However, this approach was

computationally expensive, and generated images varied with different image space regularizers. Additionally, although visually appealing, the mean squared errors (MSE) between the reconstructed and ground truth images were relatively high.

Dosovitskiy & Brox (2016) advanced feature inversion by training dedicated inverse networks for each layer of a CNN, achieving substantially higher-quality reconstructions, as reflected by lower MSE compared to ground-truth images. Applied to AlexNet (Krizhevsky et al., 2012), this approach showed that both color and spatial information are preserved across the network hierarchy. However, the method remained computationally expensive, as it required training a separate network for each layer, which becomes demanding for larger architectures. Nonetheless, their primary focus was on comparing inversion strategies rather than fully leveraging feature inversion as a tool for network interpretability.

Feature inversion should not be confused with activation maximization studies (Erhan et al., 2009; Zeiler & Fergus, 2014; Springenberg et al., 2015; Nguyen et al., 2016; Olah et al., 2017), a technique for image synthesis from intermediate network states. While feature inversion reconstructs the original input from intermediate representations to reveal the information preserved at each stage, activation maximization instead synthesizes inputs that maximally activate specific network units, such as neurons, channels, or entire layers. The resulting images typically show simple, low-level structures (e.g., edges) for early layers and more complex, high-level patterns for deeper layers.

In the broader scope of model interpretation, a variety of techniques have been developed to clarify how internal representations evolve and contribute to downstream predictions. Saliency and attribution methods, such as gradient-based saliency maps (Simonyan et al., 2014), class activation mapping (Selvaraju et al., 2017), and layer-wise relevance propagation (Bach et al., 2015) typically produce heatmaps that spatially highlight which part of an input image most influences the prediction of the model, often via gradient backpropagation or pixel-level relevance propagation. While insightful for localizing decision-critical regions, these methods do not reveal the actual visual content encoded within intermediate representations.

Perturbation-based analyses, such as occlusion sensitivity (Zeiler & Fergus, 2014) and adversarial attacks (Goodfellow et al., 2015), assess input relevance and model robustness by measuring prediction changes under controlled modifications such as masking out image regions to produce spatial relevance maps akin to saliency methods, or adding subtle adversarial noise to identify worst-case vulnerabilities. These methods are valuable to quantify sensitivity and robustness to input perturbations but offer little insight into the underlying information-processing pipeline.

Representational Similarity Analysis (RSA) methods such as singular vector canonical correlation analysis (Raghu et al., 2017), and Centered Kernel Alignment (CKA) (Kornblith et al., 2019) compare encoding spaces and quantify alignment between different layers or models to identify how representations evolve, providing both global and sample-specific similarity metrics but without directly visualizing the preserved spatial or semantic content.

Loss landscape analyses (Li et al., 2018; Garipov et al., 2018; Keskar et al., 2017) examine the geometry of the optimization surface in parameter space, revealing aspects like sharpness, flatness, and mode connectivity, to study generalization capacity and learning dynamics, yet offer little intuitive insight into the layer-wise encoding of specific inputs.

Interpretability research on TVMs, and ViT in particular, has typically employed one or more of the aforementioned broader techniques rather than feature inversion, with a primary focus on attention-based analyses, representation probing, and robustness evaluations. For instance, attribution methods such as attention rollout (Abnar & Zuidema, 2020) and relevance propagation through attention layers (Chefer et al., 2021) trace decision pathways across tokens by aggregating and propagating attention weights through successive layers, producing attention maps that aim to highlight the most influential regions or tokens for the prediction of the model. Other attribution studies assess the importance of attention heads via pruning and leave-one-out ablations, evaluating prediction changes to quantify the contribution of each head (Li et al., 2023a). Perturbation-based studies analyze robustness against occlusion, patch permutation, and adversarial noises (Naseer et al., 2021), as well as analyzing loss landscape (Park & Kim, 2022; Paul & Chen, 2022). Findings of these works indicate that ViT gradually refines its representations throughout its layers, depends

less on high-frequency features than CNNs, maintains spatial information throughout its architecture, and is particularly robust against image perturbations compared to CNNs.

RSA studies using CKA (Raghu et al., 2021) quantify alignment or separability of features across layers, also indicate that ViT has more uniform and similar representations across layers than CNNs, driven by early global information aggregation. Complementary studies (e.g., Ghiasi et al. (2022)) use activation maximization to generate synthetic patterns that most strongly activate specific ViT neurons, observing a layer-wise shift from local textures to object-level features. However, these approaches mostly highlight decision-critical regions or quantifying token/feature importance, rather than directly visualizing the full spatial and semantic content retained at intermediate stages.

In contrast to ViT, interpretability studies on TVMs for object detection, like DETR, remain limited (Chefer et al., 2021), partly because highly entangled features common for object detection are difficult to interpret with current methods. Instead, rather than analyzing existing models, recent works focus on architectural changes to improve interpretability by design, e.g., by incorporating feature disentanglement techniques or introducing new modules designed to learn prototypical features, thereby making subsequent interpretation more tractable (Yu et al., 2024; Paul et al., 2024; Rath-Manakidis et al., 2024).

To the best of our knowledge, feature inversion has not been systematically applied to to TVMs for interpretability and inverting the operations of self-attention and cross-attention layers has been considered a challenging task (Fantozzi & Naldi, 2024; Bibal et al., 2022), as these mechanisms dynamically aggregate information across all tokens, entangling spatial and semantic features in a non-local manner. Our work addresses this gap by reintroducing feature inversion as a scalable, semantically grounded interpretability tool for large-scale TVMs.

We apply feature inversion to the two TVMs DETR (Carion et al., 2020), designed for object detection, and ViT (Dosovitskiy et al., 2020), designed for image classification. DETR comprises a convolutional backbone, a transformer encoder-decoder, and a multi-layer perceptron (MLP) prediction head. On the other hand, ViT features a linear projection layer (serving as its backbone) followed by a transformer encoder and a MLP head. In both architectures, images are represented as sequences of tokens within their encoders, maintaining a one-to-one correspondence with their spatial locations, making them well-suited for feature inversion. We additionally advance the feature inversion approach by training modular inverse components locally, eliminating the need to train separate networks for each layer. This advancement greatly improves efficiency and scalability, enabling the application of it to large-scale architectures.

We leverage our novel modular feature inversion approach as a systematic interpretability framework, and instead of inferring importance from gradients, perturbations, or similarity metrics, we directly reconstruct images from intermediate representations, providing human-interpretable depictions of what each stage encodes. This enables both qualitative inspection of visual richness, abstraction level, and spatial fidelity, as well as quantitative analysis, e.g., reconstruction error comparison, across stages and models. Furthermore, using targeted perturbations into feature space, we enable step-by-step analysis of the model's information processing pipeline and robustness. Moreover, the framework serves as a diagnostic tool for detection failures, offering a detailed view into intermediate representations when prediction errors occur.

## 3 Methods

### 3.1 Feature inversion

Feature inversion attempts to reconstruct input images from intermediate representations within a neural network to gain insights into and interpret the processing mechanisms of the network. To formalize the method, let $\mathcal{N} : \mathcal{X}_0 \to \mathcal{X}_L$ be a neural network with parameters $\theta$, mapping from an image space $\mathcal{X}_0$ to an output space $\mathcal{X}_L$. Assume that $\mathcal{N}$ consists of $n$ processing stages of interest indexed by $\mathcal{P} := (1, \ldots, n)$, a subset of the layers of the network. We denote the representation of an image at input stage as $\mathbf{x}_0 \in \mathcal{X}_0$ and its representation at stage $j \in \mathcal{P}$ as $\mathbf{x}_j := \mathcal{N}_{0:j}(\mathbf{x}_0; \theta_{0:j})$, where $\mathcal{N}_{i:j} : \mathcal{X}_i \to \mathcal{X}_j$ represents the component of the network spanning layers $i$ to $j$ with parameters $\theta_{i:j} \subseteq \theta$.

Furthermore, we define the approximate inverse of component $\mathcal{N}_{i:j}$ as the neural network $\mathcal{N}_{j:i}^{-1} : \mathcal{X}_j \to \mathcal{X}_i$ with parameters $\phi_{j:i}$. The reconstruction of $\mathbf{x}_i$ from processing stage $j \in \mathcal{P}$ is given by $\hat{\mathbf{x}}_{j:i} := \mathcal{N}_{j:i}^{-1}(\mathbf{x}_j)$. When $i = 0$, we refer to the reconstruction as image reconstruction and layer reconstruction otherwise.

Feature inversion for the network interpretability approach by Dosovitskiy & Brox (2016) follows two steps. First, separate inverse components $\mathcal{N}_{j:0}^{-1}$ are trained for each $j \in \mathcal{P}$ by minimizing the expected mean squared error (MSE) between image reconstructions $\hat{\mathbf{x}}_{j:0}$ and their corresponding input images $\mathbf{x}_0$. Then, the inverse components are used to generate reconstructed images from the various processing stages, enabling an interpretation of the processing mechanisms of the forward network.

Intuitively, feature inversion relies on the principle that inverse components reconstruct an image by generating an average over plausible images corresponding to a given representation $\mathbf{x}_j$. As a network processes an input, it abstracts information and omits details at various stages. If the inverse components are sufficiently powerful, the reconstructed image will reflect these abstractions and omissions at the pixel level. Analyzing these pixel-level transformations allows for an assessment of what information is retained, omitted, or abstracted at different processing stages, thereby enhancing the interpretability of the network.

## 3.2   Modular feature inversion

We modify the classic approach to feature inversion in a key aspect. Instead of training inverse components to map from $\mathcal{X}_j$ to $\mathcal{X}_0$ directly, we train local inverse components, learning $\mathcal{N}_{j:j-1}^{-1}$ for all $j \in \mathcal{P}$. Formally, given a training set of images, we optimize the expected MSE for an inverse component over parameters $\phi_{j:j-1}$:

$$L_{\text{MSE}}(\phi_{j:j-1}) := \mathbb{E}\left[\|\mathbf{x}_j - \mathcal{N}_{j:j-1}^{-1}(\mathbf{x}_j)\|_2^2\right] \tag{1}$$

With the modular approach, we then obtain image reconstructions by sequentially applying the trained inverse components from any processing stage $b \in \mathcal{P}$:

$$\hat{\mathbf{x}}_{0:j} := (\mathcal{N}_{1:0}^{-1} \circ \cdots \circ \mathcal{N}_{j:j-1}^{-1})(\mathbf{x}_j) \tag{2}$$

Our modular approach offers various advantages. Biasing the inverse solution space of inverse components by independently training inverse components maintains greater symmetry between the processing stages of the forward and backward path. Additionally, computational efficiency is greatly improved, as fewer, smaller components are required compared to training separate inverse components attempting to inverse the entire forward path for each stage.

This efficiency can be illustrated by comparing the total number of trainable parameters in the full-path approach of Dosovitskiy & Brox (2016) versus our modular approach: Let $\mathcal{N}$ be a DNN with $p$ parameters and $n$ processing stages of interest, for simplicity, each with $p/n$ parameters. In the full-path approach, $n$ inverse models are trained, each inverting an increasingly larger network portion. Assuming each inverse network roughly mirrors its forward path, the total parameter count for all inverse networks is $\sum_{i=1}^{n} i \cdot \frac{p}{n} = \frac{np+p}{2}$, scaling linearly with $n$. In contrast, for the same $\mathcal{N}$, our modular method uses $n$ modular inverse components of size $p/n$, totaling $p$ parameters, constant in $p$ and independent of $n$.

## 3.3   Application to DETR and ViT

We applied modular feature inversion to the two TVMs, DETR and ViT (see Figure 2 for an illustration of our approach on DETR). The procedure for ViT follows the same methodology. Specifically, we analyzed pretrained DETR-R50 and ViT-B/16 as they are a reasonable compromise between performance and size. For DETR, we identified four processing stages of interest $\mathcal{P} := (1, 2, 3, 4)$ while for ViT, we considered two stages $\mathcal{P} := (1, 2)$. These correspond to the representations after processing by the backbone (bb), encoder (enc), decoder (dec), and prediction head (pred). Note that ViT does not consist of a decoder, and we excluded the prediction head of ViT from our analysis, as it operates on a single token, discarding the remaining encoder sequence. In contrast, the prediction head of DETR processes the entire decoder sequence.

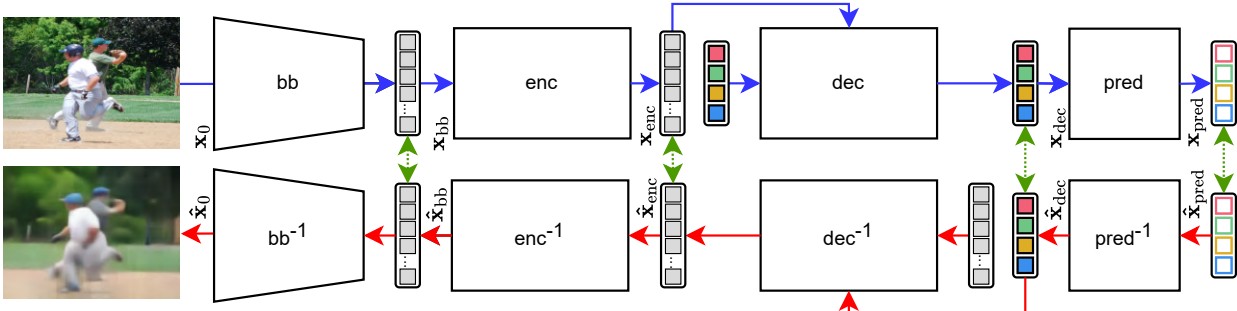

Figure 2: Modular feature inversion of DETR. The top half illustrates the main components of DETR, with blue arrows indicating the forward path of an input image through the architecture. The bottom half shows our modular inversion approach, where each component is inverted individually. Red arrows trace the backward path from predictions through the inverse components, enabling image reconstruction from any processing stage. Green double-arrows indicate correspondence between representations, i.e., representations output by a respective forward component can be used as input to the corresponding inverse component during inference or as a supervision signal during training.

To improve readability, we will henceforth write bb instead of $\mathcal{N}_{0:1}$, bb$^{-1}$ instead of $\mathcal{N}_{1:0}^{-1}$, and $\mathbf{x}_{bb}$ instead of $\mathbf{x}_1$, with analogous notation for other components, e.g., enc for $\mathcal{N}_{1:2}$. When referring to an image reconstruction from a specific processing stage, such as the encoder, we write $\hat{\mathbf{x}}_{enc:0}$.

Our inverse components were designed to mirror their respective forward components. For DETR, we implemented a deconvolutional network as bb$^{-1}$, resembling an inversion of DETR's backbone (ResNet-50 (He et al., 2016)). We set enc$^{-1}$ to be structurally equivalent to enc, but with swapped inputs and outputs. Similarly, we defined dec$^{-1}$ as structurally equivalent to dec, but initialized its input as blank tokens that self-attend to each other and cross-attend to $\mathbf{x}_{dec}$. For pred$^{-1}$, we used a simple MLP that takes the concatenation of bounding box and the full distribution of class logits as input.

For ViT, we employed a local small deconvolutional network as bb$^{-1}$. Although bb in ViT is a simple, local invertible linear transformation, its analytical inverse proved to be ill-conditioned, making it highly sensitive to layer reconstruction errors between $\mathbf{x}_{bb}$ and $\hat{\mathbf{x}}_{enc:bb}$. For enc$^{-1}$ we used a structurally equivalent component to enc, but reversed the direction of information flow. A detailed description of our components is provided in the supplementary material and our code repository[1].

We trained all inverse components on the COCO 2017 (Fleet et al., 2014) training dataset and evaluated image reconstructions on the corresponding test set. We trained at least three instances for each inverse component variant and interchanged them during analysis to ensure that findings are not attributable to random variation.

## 4 Results

### 4.1 Assessing modular approach

After training the inverse components for DETR and ViT, we first tested the feasibility of our modular feature inversion approach. To this end, we began by evaluating reconstructed images of arbitrary examples generated with our method from the various processing stages of the two architectures.

Figure 3 presents exemplary image reconstructions. Early-stage image reconstructions from both DETR and ViT maintain the overall scene layout and coarse object structure. However, ViT preserved fine-grained details more accurately, while DETR reconstructions show early signs of blurring even at the backbone stage. These differences become more pronounced in deeper stages, where DETR reconstructions progressively lose

---

[1]code submitted in supplement to ensure anonymity during double-blind peer review

structural detail, introduce color shifts, and abstract away background elements, suggesting a systematic abstraction process. In contrast, ViT reconstructions remain comparatively faithful across stages.

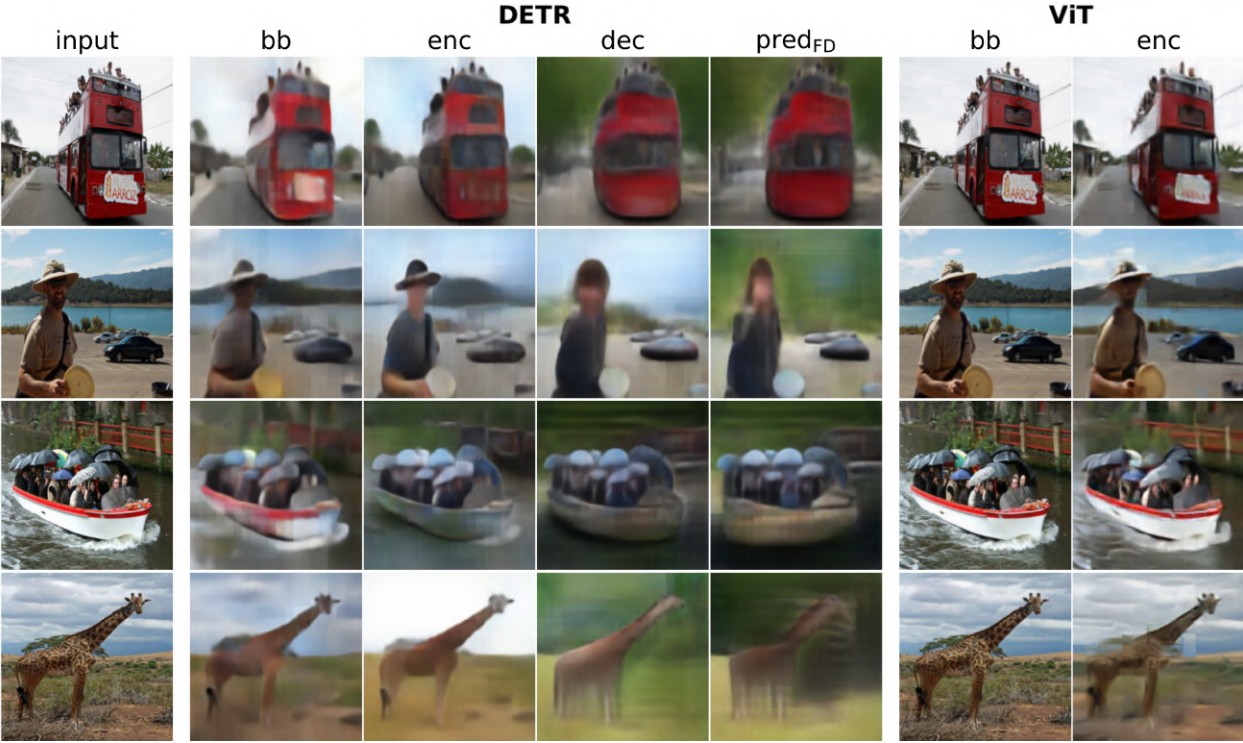

Figure 3: Image reconstructions from processing stages. Column 1 shows the original input images. Columns 2-5 and 6-7 show reconstructions from different processing stages of DETR and ViT, respectively.

We quantified these discrepancies between image reconstructions in DETR and ViT in Figure 4 (left), using the average MSE across different processing stages. As expected, reconstruction error increases at later stages in both models; however, ViT maintains significantly lower MSE than DETR throughout, consistent with our first assessment of reconstructed images. Notably, in DETR, the MSE from the decoder stage onward exceeds the baseline error of comparing each image to the dataset mean (a grayish, structureless reference image). While this could superficially suggest that representations from the decoder stage onward are less informative than a simple average image, visual inspection of the corresponding reconstructions in Figure 3 reveals the contrary: Despite the higher reconstruction error, these outputs preserve structured, object-specific content, indicating that the underlying representations encode nontrivial scene information.

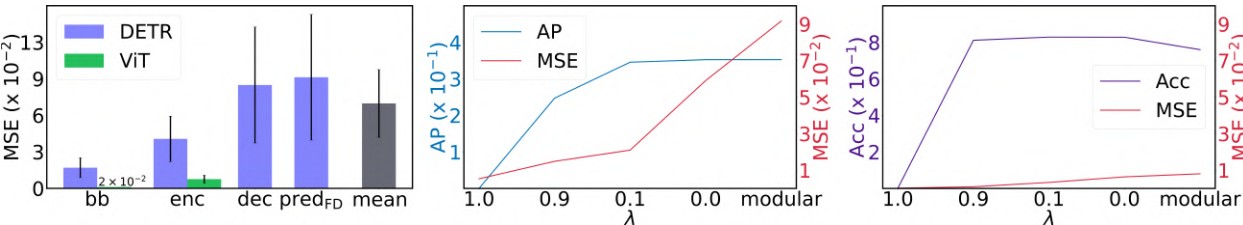

Figure 4: **Left**: Average reconstruction error across processing stages for DETR and ViT on the COCO validation dataset. "mean" denotes the average reconstruction error between validation images and the mean image of the dataset. **Center**: Reconstruction loss (MSE) at the decoder stage versus object detection performance (AP) for DETR, evaluated across different values of $\lambda$. The label "modular" indicates the performance of our modular feature inversion approach without fine-tuning. **Right**: Reconstruction loss (MSE) at the encoder stage versus classification accuracy (Acc) for ViT across varying $\lambda$.

This observation suggests a desirable property of our modular approach to feature inversion: Despite the potential for error accumulation across sequential inverse modules, the approach enables reconstructions that retain stage-specific transformations without collapsing into a globally averaged output, even when applied sequentially across deep model stages. This hypothesis prompted a more systematic evaluation of the validity of our modular feature inversion framework presented in Section 4.2.

## 4.2 Validating modular approach

We validated our modular approach to feature inversion by comparing the image reconstructions obtained with our method to those generated using the classic feature inversion approach. For classic feature inversion, we selected a processing stage of interest for DETR and ViT, specifically, $\mathbf{x}_{\text{dec}}$ for DETR and $\mathbf{x}_{\text{enc}}$ for ViT, and trained networks to reconstruct input images directly, following the classic approach to feature inversion. That is, we trained inverse networks to determine the optimal parameters $\phi_{\text{dec}:0}$ for DETR and $\phi_{\text{enc}:0}$ for ViT.

As an additional control experiment , we also fine-tuned the forward weights of both models by incorporating an image reconstruction loss. This allowed us to assess whether the loss of detail in reconstructions was due to limitations in the forward components rather than ineffective inverse components. Specifically, we combined the reconstruction loss with the objectives of the respective architectures, $L_{\text{OBJ}}$, following the approach of Rathjens & Wiskott (2024):

$$L(\theta \cup \phi_{j:0}) = \lambda L_{\text{MSE}}(\theta) + (1 - \lambda)L_{\text{OBJ}}(\theta) + L_{\text{MSE}}(\phi_{j:0}) \tag{3}$$

We trained four model variants with $\lambda \in \{0.0, 0.1, 0.9, 1.0\}$, setting $j$ to dec for DETR and enc for ViT. Notably, $\lambda = 0.0$ corresponds to the classic feature inversion approach, where the forward weights $\theta$ are not influenced by the reconstructions loss, while the other $\lambda$-values correspond to fine-tuned versions of our models. For ViT, we used the ImageNet-1K dataset (Krizhevsky et al., 2012), as ViT is designed for classification, not object detection, making the COCO object detection dataset less suitable for fine-tuning..

Figure 5 presents the image reconstructions obtained with fine-tuned inverse models alongside those generated using our modular approach and the classic feature inversion approach. Across all examples, a consistent pattern emerges: for high $\lambda$ values, the reconstructions maintain high fidelity, capturing image details accurately. As $\lambda$ decreases, reconstruction quality deteriorates. This effect is particularly pronounced in DETR, especially in the last two columns, where blur is high. In contrast, ViT exhibits this effect to a lesser degree. Interestingly, for DETR, clear differences emerge between the last two columns: The reconstructions in the $\lambda = 0$ column, which represent the classic feature inversion approach, exhibit a grayish tone, whereas those in the modular approach column display more saturated colors.

We quantitatively analyzed this pattern in Figure 4, which displays the mean squared error (MSE) alongside average precision (AP) for DETR (center plot) and MSE alongside accuracy for ViT (right plot). The results align with the qualitative assessment of the reconstructed images: As $\lambda$ decreases, reconstruction error increases. Moreover, as $\lambda$ decreases, the object detection and classification performances of DETR and ViT improve, highlighting a trade-off between reconstruction quality and the tasks of the architectures. Unsurprisingly, the MSE is lower for the classic feature inversion approach than for our modular approach. As a side note, despite this trade-off, our fine-tuned ViT for $\lambda$ set to 0.0, 0.1, and even 0.9, achieved slightly higher validation accuracy on ImageNet-1K than the released ViT-B/16 model (used in our modular approach).

We interpret these findings as follows. Firstly, incorporating reconstruction-driven information into the forward path enhances reconstruction quality but degrades the performance of the architectures, in line with previous work (Rathjens & Wiskott, 2024). This trade-off suggests that blur in reconstructions is not a result of ineffective inverse components but rather a consequence of how information is processed in the forward model. Importantly, this implies that poor reconstructions are not detrimental to interpretability; on the contrary, they highlight which information is omitted or abstracted at different stages of the model. Secondly, our modular approach to feature inversion enhances reconstruction properties for interpretability. While the classic approach has a better reconstruction performance in terms of MSE, it does so by shifting

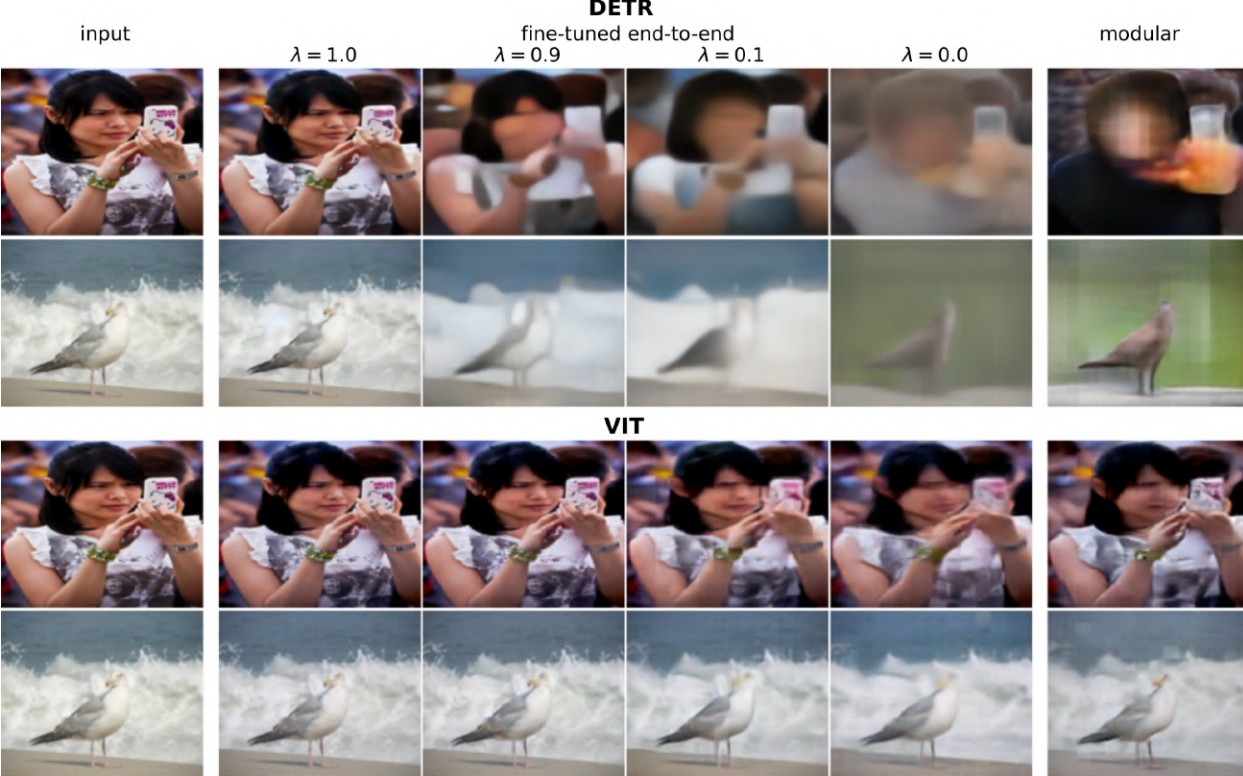

Figure 5: Reconstructions with fine-tuned models. Columns 2–4 show reconstructions from models fine-tuned with different $\lambda$ values. Column 1 displays input images, and column 6 shows reconstructions using our modular feature inversion approach. Top: Images from the DETR dec processing stage. Bottom: Images from the ViT enc processing stage.

colors toward a grayish tone, reflecting the average color of the COCO dataset, particularly evident for DETR. This shift occurs because the inverse network in the classic approach can exploit global dataset-wide color statistics as they are not trained locally. In contrast, the modular approach better preserves stage-specific information, though the MSE between input and reconstructed images may increase, particularly when reconstructed from deeper representations.

## 4.3   Analyzing color

Having established the feasibility and validity of our modular inversion framework, we turned to its primary purpose, i.e., interpreting intermediate representations in DETR and ViT.

Building on our initial observations suggesting substantial differences in color processing between DETR and ViT, we conducted a systematic analysis to explore these differences in more detail. Precisely, we recolored specific objects in input images and evaluated the influence of these modifications on image reconstructions from the various processing stages. For recoloring, we used the segmentation annotations from the COCO dataset to apply six different color filters in the HSV color space to various object categories. Specifically, we adjusted the hue of specified objects to red, green, blue, shifted the hue values by 120 or 240 degrees, or converted all image pixels to grayscale. Figure 6 illustrates recolored images alongside their corresponding reconstructions for several examples. Each row presents an example from a different object category with a distinct color filter applied. Each column presents a reconstruction from a different processing stage.

For DETR, we observe that color perturbations are preserved in image reconstructions from the backbone representations for all objects and filters but gradually fade or disappear almost entirely in image reconstructions from the encoder representations. In $\hat{\mathbf{x}}_{\text{dec:0}}$ practically no color perturbation remains. Instead, colors

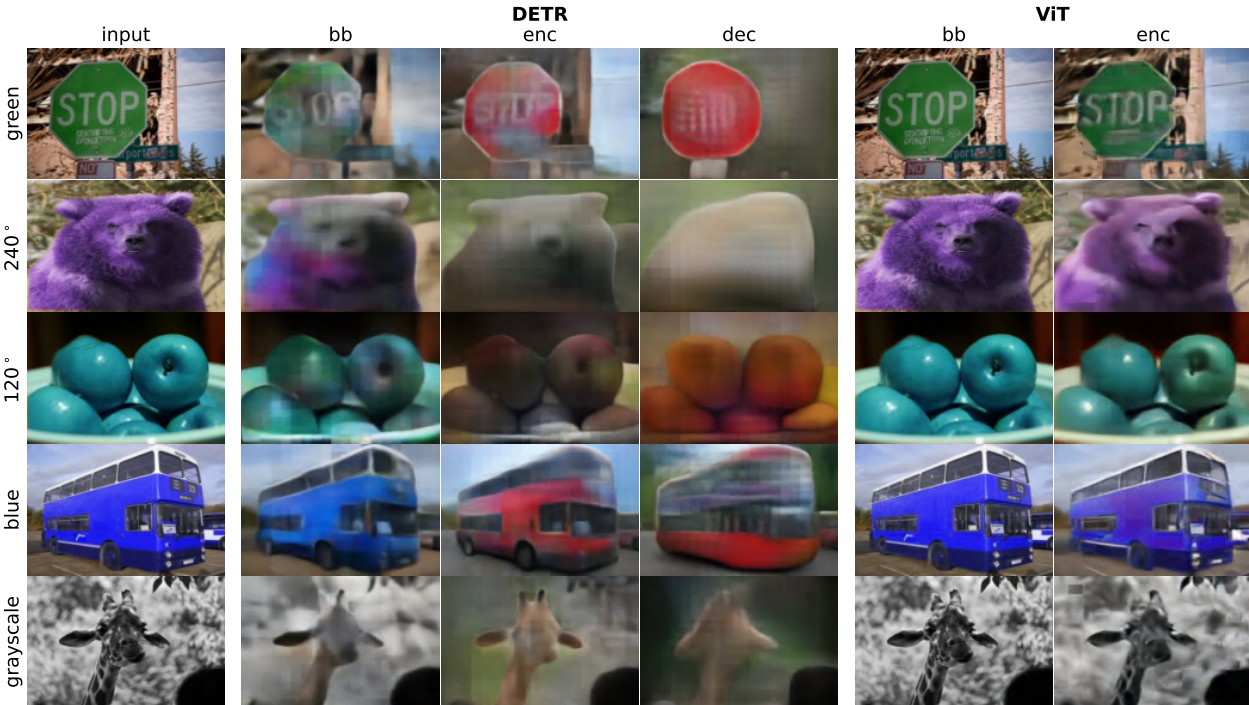

Figure 6: Effects of color perturbations. Rows show images where specific object categories were color-perturbed (from top to bottom: stop sign colored green, bear with colors rotated by 240°, apple with colors rotated by 120°, bus colored blue, giraffe converted to grayscale). Columns 2-4 and 5-6 show reconstructions from different processing stages of DETR and ViT, respectively.

shift toward prototypical representations (red for the stop sign and bus, brown for the bear, red or yellow for the apples, and yellow for the giraffe) even when color information was deleted (see giraffe). In contrast, we do not observe a similar effect in ViT, as color perturbations remain visible in image reconstructions from all processing stages.

We quantified the response to color perturbations by computing the average pairwise MSE between image reconstructions of differently perturbed images from each processing stage. Specifically, given an image $\mathbf{x}_0$, we applied each color filter separately, generating six perturbed versions. We calculated the average pairwise MSE between these perturbed images at the input stage. Similarly, for the backbone stage, we computed the average pairwise MSE between the six corresponding reconstructed images $\hat{\mathbf{x}}_{bb:0}$, following the same approach for the encoder and decoder stages. The left plot in Figure 7 presents these MSE values, averaged across all categories and images in the dataset.

For DETR, we observe that the average pairwise MSE decreases progressively from $\mathbf{x}_0$ to $\hat{\mathbf{x}}_{enc:0}$, indicating increasing similarity. However, at the decoder stage, the MSE returns to input levels. This observation aligns with our qualitative analysis, confirming that reconstructions tend to converge to the same or similar colors as they progress through the DETR architecture. The increase in average pairwise MSE at the decoder stage is likely not due to color divergence but rather distortions in object shapes. For ViT, the preservation of color perturbations throughout the architecture is reflected in an almost constant pairwise MSE across processing stages.

The greater loss of color information in DETR compared to ViT suggests that DETR is more robust to color changes than ViT. We tested this hypothesis by evaluating the performance of each architecture on recolored images (see right plot Figure 7). Specifically, we recolored entire images from the ImageNet dataset and measured classification performance for ViT, as segmentation data was not available. To ensure a fair comparison, we also applied full-image recoloring for DETR. The results show that accuracy of ViT drops compared to the default setting, whereas DETR remains unaffected.

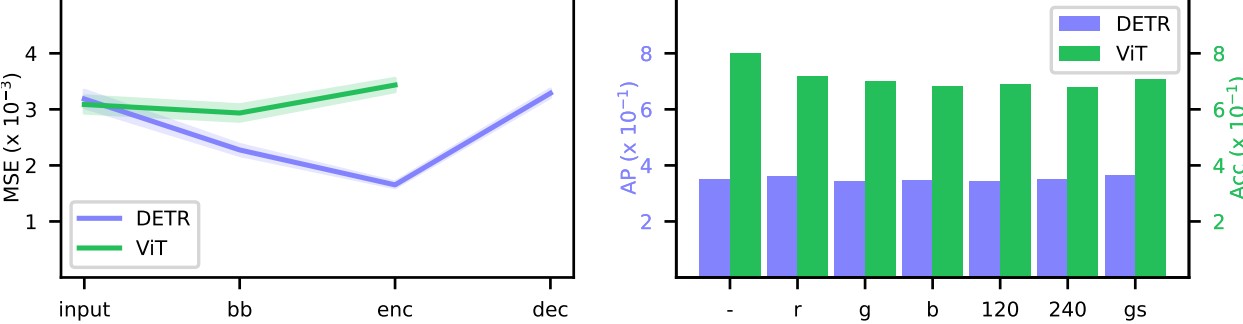

Figure 7: Quantification of color perturbations. Left: Average pairwise MSE between image reconstructions of differently perturbed versions of an image, comparing inputs and reconstructions across processing stages. Shaded area indicates 95% confidence intervals over the COCO test set. Right: DETR's and ViT's sensitivity to color perturbations (none, red, green, blue, 120° shift, 240° shift, grayscale) in relation to the performance of their objectives.

### 4.4 Analyzing structure

Our initial assessment of reconstructed images indicated that DETR progressively alters the image structure across its processing stages, whereas ViT tends to preserve it. To investigate this phenomenon in greater depth, we again reconstructed images from various stages of both architectures, this time focusing on the analysis of structural changes. Given the particularly interesting behavior observed in DETR, we employed two variants for its $\text{pred}^{-1}$: a standard $\text{pred}_{\text{FD}}^{-1}$, which receives $\mathbf{x}_{\text{pred}}$ with full distribution of class logits as input, and an additional $\text{pred}_{\text{OH}}^{-1}$, which takes a one-hot encoded variant of $\mathbf{x}_{\text{pred}}$. The latter retains only the highest-confidence class per detected object, discarding information about the uncertainty in class predictions. Bounding boxes are retained in both variants. We hypothesized that the full distribution of class confidences, beyond just the top prediction, encodes meaningful visual cues. The one-hot variant allowed us to evaluate more prototypical reconstructions of certain objects, without the model being able to exploit the uncertainty information and low-confidence class associations during the reconstruction process.

Figure 8 displays exemplary image reconstructions. For DETR, we observed that low-level structural information is generally well-preserved in reconstructions from $\mathbf{x}_{\text{bb}}$. Notably, at the later stages starting from dec, objects undergo significant alterations, including changes in size, shape, structure and orientation (e.g., the person in the first row appears taller with a lowered hand, the sunflowers in the third row shift into a generic green plant, and the horse in the fourth row is reoriented to face right), the addition of contextual elements (left person in the second row appears to be wearing a suit in reconstructions from dec and pred, inferred from the presence of a tie), or complete omissions of objects (e.g. the bollards in the second row, or the photo frame on the wall in the third row are completely abstracted out). Furthermore, we observe some artifacts, e.g., in the reconstructions from $\text{pred}_{\text{OH}}$ in the fourth row, a dark object appears near the horse that seems to be another person, which is not present in earlier stages. These transformations appeared repeatedly across diverse samples and object classes, suggesting that the model learns structured abstraction behaviors that are consistent within each class.

In contrast, ViT reconstructions show little structural change across stages. Object shape, spatial configuration, and contextual elements are consistently preserved, suggesting that ViT retains low-level visual and semantic information without applying the same degree of abstraction observed in DETR.

We interpret these observations as follows. In higher processing stages, DETR tends to omit image details that are not relevant to object detection, such as objects that are not explicitly recognized (e.g., the omission of bollards and photo frame, compared to detected objects indicated by bounding boxes in the input images). Instead of preserving raw image details, DETR represents objects in a prototypical manner, discarding information deemed irrelevant for recognition, such as pose and shape variations or orientation changes (e.g., the altered posture of the person or the transformed sunflowers). Additionally, DETR appears to learn priors about object co-occurrences and typical scene compositions. It may modify contextual elements to enhance

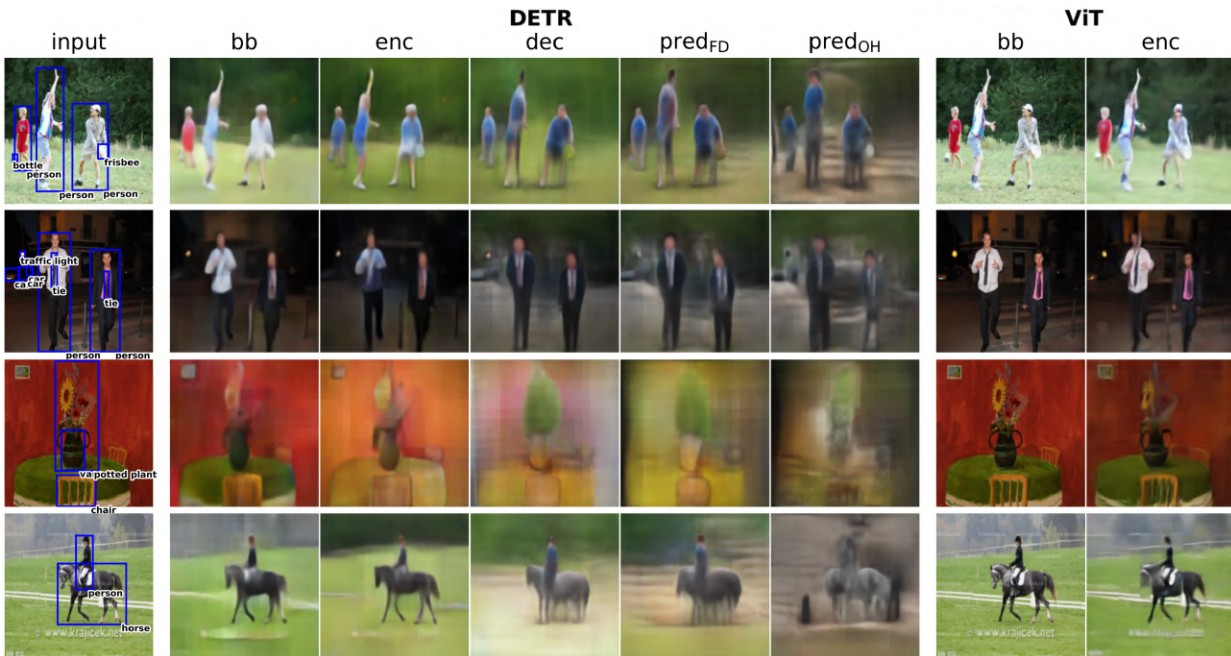

Figure 8: Structural transformations analysis in DETR and ViT

object recognizability, as seen in the addition of a suit coat to emphasize the tie. Using only top-scoring classes for reconstructions can also lead to semantically relevant hallucinations, like a person appearing near a horse, emphasizing the role of model confidence in activating the co-occurrence of related objects. On the other hand, ViT does not appear to undergo these abstractions, as image details remain largely preserved throughout its architecture.

## 4.5 Analyzing spatial correlations

Throughout our experiments, we observed that ViT preserves image details more effectively across processing stages than DETR, suggesting a stronger spatial correspondence between the input image and its token representations. To further examine this spatial correlation, we replaced 20% of randomly selected tokens at two processing stages with identical uniform noise added to the respective positional embeddings, and analyzed the resulting image reconstructions. Specifically, we manipulated $\mathbf{x}_{bb}$ and generated reconstructions $\mathcal{N}^{-1}_{bb:0}(\mathbf{x}_{bb})$ and $\mathcal{N}^{-1}_{enc:0}(\mathcal{N}_{bb:enc}(\mathbf{x}_{bb}))$, which we refer to as $bb_{man}$ and $enc_{man+}$. Additionally, we manipulated $\mathbf{x}_{enc}$ to generate reconstruction $\mathcal{N}^{-1}_{enc:0}(\mathbf{x}_{enc})$ which we refer to as $enc_{man}$. Note that positional information of tokens from the encoder stage of ViT cannot be preserved, as explicit positional encodings are no longer present at that processing stage.

Our reasoning for the experimental setup was as follows: If the manipulation of tokens result in visible distortions at their corresponding spatial locations, while the rest of the image remains unchanged, it would indicate that the reconstruction process relies primarily on local features. Such behavior suggests a low level of abstraction, where image information is not distributed across all tokens. Conversely, if the manipulated tokens do not produce localized distortions but instead affect the entire reconstruction, it implies that the image information is distributed across tokens, pointing to a higher level of abstraction and reduced spatial locality.

To enable a more isolated analysis of enc, we introduced a local inverse backbone variant for DETR. In this configuration, each image patch is reconstructed solely from the token corresponding to its spatial location, without incorporating global features. This design decouples the global integration of image details across tokens performed by the encoder from that of the backbone.

Figure 9 shows results for two example images for both architectures. For the DETR setup with the standard $bb^{-1}$, token manipulation leads to slightly increased blurring and color shifts in the reconstructions across all processing stages. However, the reconstructions do not reveal which tokens were manipulated, as the noisy tokens were consistently filled in with plausible content.

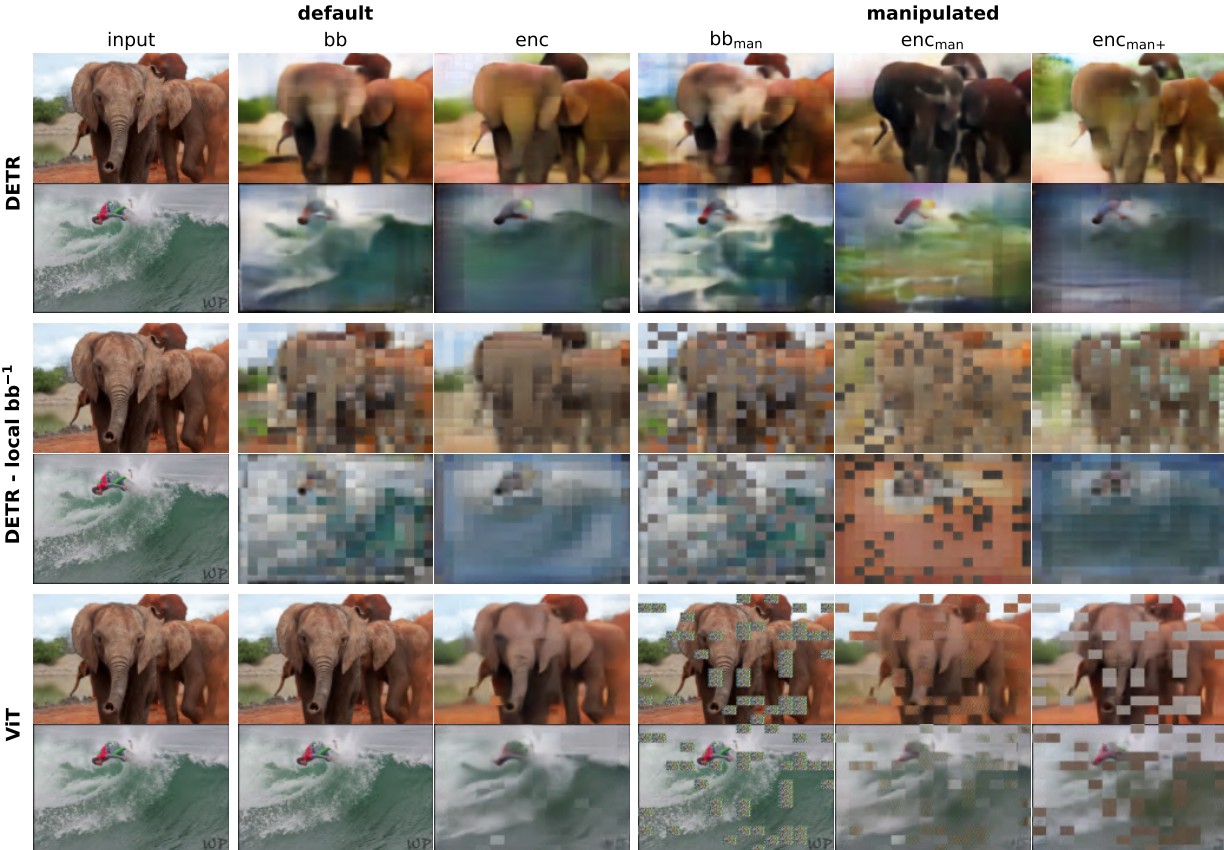

Figure 9: Image reconstructions from default (unmanipulated) and manipulated representations with DETR, a DETR variant with a local inverse backbone, and ViT. Tokens were replaced with the same random uniform noise at different processing stages before reconstruction. For $enc_{man+}$, embeddings were manipulated at the backbone stage, processed through enc, and then used for reconstruction.

With the local inverse backbone variant for DETR, overall reconstruction quality deteriorates significantly, as expected. Unlike the standard inverse backbone, reconstructed images with the local version enable a more accurate identification of the manipulated tokens. Since the local $bb^{-1}$ reconstructs each image patch using only a single token, and all manipulated tokens are replaced with identical noise, the corresponding patches appear visually identical, as visible in the $bb_{man}$ setup.

In the $enc_{man}$ setup, manipulated tokens can still be identified, as their corresponding reconstructed patches differ from those based on unmanipulated tokens. However, these differences are less pronounced, suggesting that manipulated tokens integrate some contextual information from surrounding tokens. In the $enc_{man+}$ condition, patches reconstructed from manipulated tokens are nearly indistinguishable from others, as the noisy tokens blend seamlessly into the overall image.

The appearance of reconstructed images from manipulated representations in DETR stands in sharp contrast to those obtained from ViT. For ViT, manipulated tokens manifest as visible noise within their corresponding patches, while unmanipulated patches remain unaffected.

The differences in image reconstructions from manipulated representations strongly support the hypothesized distinction in information processing between the two architectures. While both the DETR backbone and

encoder distribute image details associated with a given location across multiple tokens, ViT components preserve a spatial correspondence between tokens and image locations. As a result, the inverse components in ViT do not require global integration to reconstruct the image, an effect particularly evident in the enc$_{man+}$ setup, where the manipulated tokens remain clearly identifiable despite being processed through multiple stages.

The lower level of abstraction in ViTs suggests that, in the forward pass, greater emphasis is placed on attention from the class token to image tokens rather than self-attention among image tokens. To test hypothesis, we disabled self-attention in enc retaining only the cross-attention from the class token to image tokens, and fine-tuned the architecture on ImageNet-1K. Remarkably, this modified model still achieved approximately 69% top-1 accuracy, only seven percentage points lower than the 76% accuracy of ViT-B/16 with enabled self-attention, supporting our hypothesis.

## 4.6    Analyzing detection errors in DETR

Our method also enables a visual inspection of detection errors by examining the reconstructed images to find out how DETR encodes, or fails to encode, objects across stages. As illustrated in Figure 10, objects that are ultimately not detected (e.g., the bicycle in the second row or the potted plants in the third row) are gradually suppressed across the processing stages. Although clearly visible in the input, these elements begin to fade in the reconstructions from bb and enc, and leave no trace in the reconstructions from dec or pred. This gradual disappearance suggests that the model deems them irrelevant and filters them out during object query formation or matching.

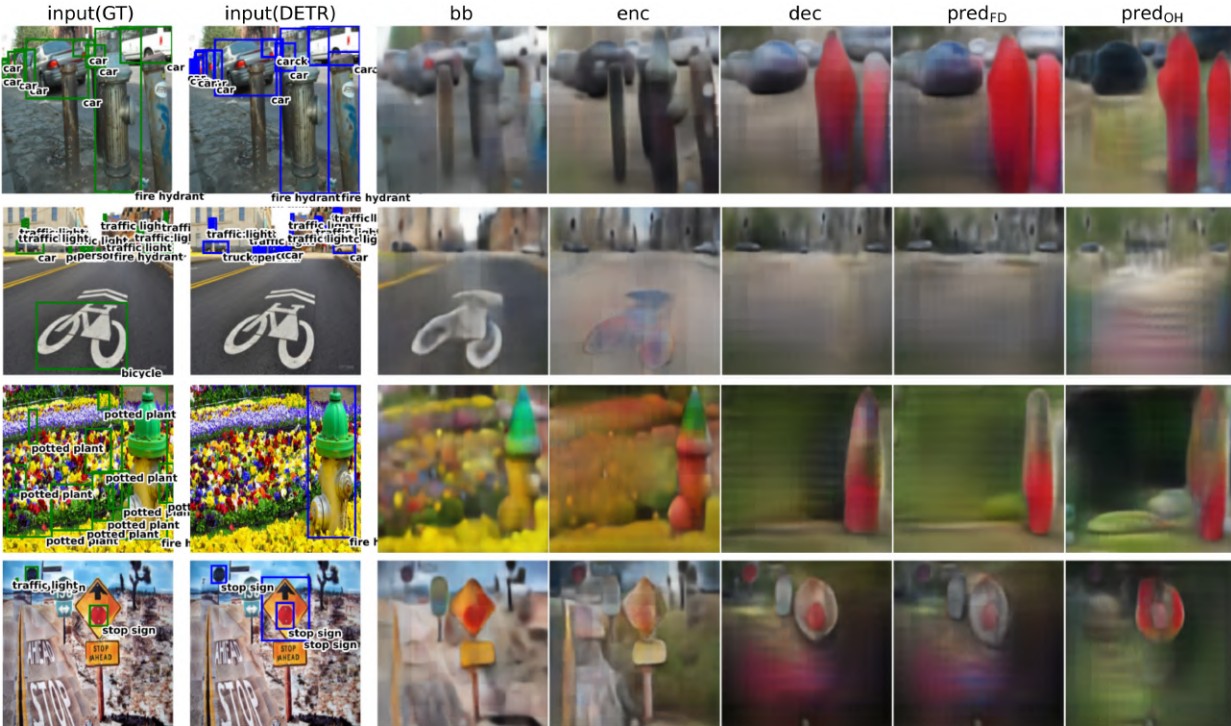

Figure 10: Image reconstructions from various processing stages of DETR. Analysing detection errors using reconstructions of various stages. The first and second columns depict input images along with ground truth labels and predictions of DETR, respectively.

In contrast, false positives often exhibit the opposite behavior: Reconstructions from later stages reveal a shift toward features associated with incorrect classes (e.g., the second fire hydrant in the first row or the second stop sign in the fourth row). This suggests that, if DETR misinterprets certain features or contextual cues, it constructs coherent features and consolidates them into prototypical object representations.

These observations provide a visual trail of where detection errors arise by revealing the stages in the processing pipeline where critical information is lost or misrepresented. This stage-wise visual access to internal representations makes reconstruction-based analysis a valuable diagnostic tool for interpreting the inner workings of DETR, highlighting where in the architecture corrective refinements might be most effective.

### 4.7  Analyzing intermediate layers

Until now, we have applied feature inversion only to representations from selected processing stages, thereby excluding layers not explicitly chosen for our analysis. However, unlike in CNNs, transformer-based vision models offer a unique analytical opportunity: The intermediate representations within both the encoder and decoder maintain a consistent shape across layers. This property allows intermediate encoder representations to be passed through the inverse backbone and inverse encoder, and intermediate decoder representations through the inverse decoder, even though these components were not trained for this purpose. Leveraging this unique property, we explored whether our inverse components could reconstruct images from intermediate encoder and decoder representations, despite the mismatch in training context. Specifically, we analyzed intermediate representations from the encoder and decoder of DETR, as well as from the encoder of ViT. Figure 11 provides illustrative examples.

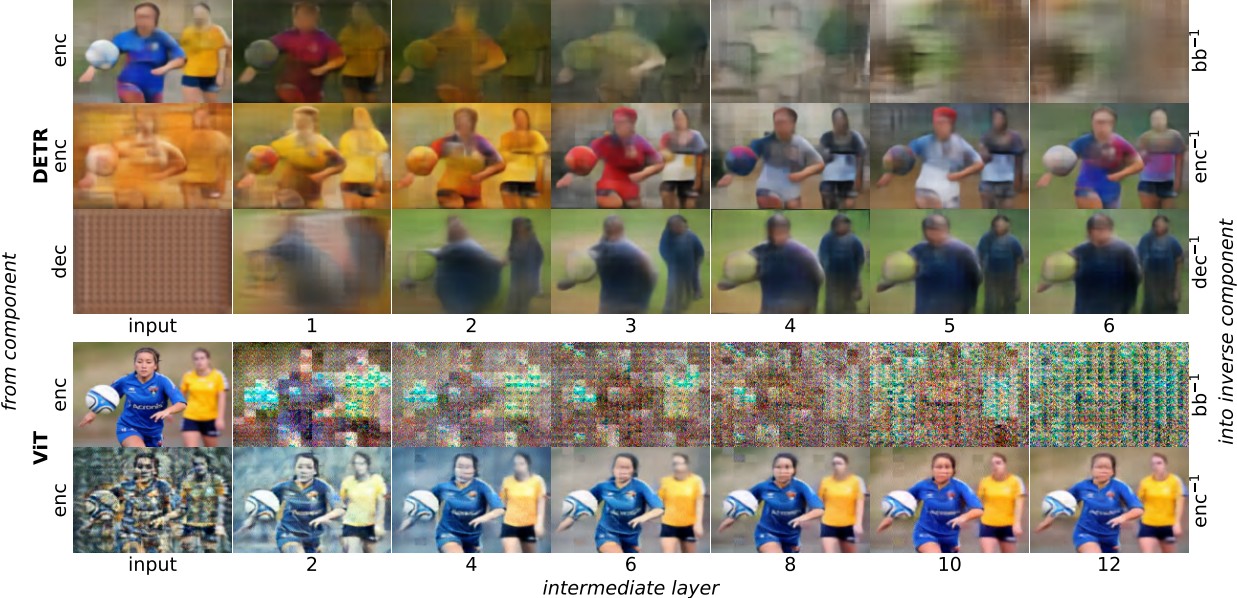

Figure 11: Image reconstructions from intermediate encoder and decoder layers. The left y-axis labels indicate the component from which we extracted the respective intermediate representation. The x-axis labels denote the specific intermediate layer corresponding to the intermediate representation. The right y-axis labels indicate into which inverse component we fed the intermediate representation.

Predictably, for intermediate representations of both architectures, we obtained best reconstruction performances for the representations the inverse components were trained on: $\mathbf{x}_{bb}$ for $bb^{-1}$ , $\mathbf{x}_{enc}$ for $enc^{-1}$ and, for DETR, $\mathbf{x}_{dec}$ for $dec^{-1}$. The quality of reconstructions gradually decreases as we move farther away from the representations the inverse components were trained on, a pattern particularly evident for the input to $dec^{-1}$ since decoder tokens initially hold values that are independent of the input image.

Despite of this degradation, image features are generally preserved across intermediate layers, especially when feeding intermediate encoder representations into $enc^{-1}$. For DETR, most variations in $bb^{-1}$ and $enc^{-1}$ appear as color shifts, whereas reconstructions from $dec^{-1}$ exhibit greater stability in color than in shape. For ViT, reconstructions from $enc^{-1}$ preserve both shape and color, while $bb^{-1}$ display strong tiling effects, likely due to the local operations of the inverse backbone. Nevertheless, both color and shape remain discernible.The overall stability of reconstructions across layers is noteworthy, as inverse modules might be

expected to produce only noisy outputs when applied to intermediate embeddings they have not been trained on.

From these observations, we draw three key conclusions. Firstly, the difference in feature preservation between DETR and ViT further highlights their distinct approaches to information abstraction, as DETR progressively alters colors throughout its hierarchy. Secondly, intermediate embeddings in transformer-based vision models evolve gradually across layers, as suggested by Raghu et al. for ViTs (Raghu et al., 2021) and by Liu et al. for LLMs (Liu et al., 2023). Finally, feature inversion is particularly well-suited for TVMs, as inverse components can be applied across multiple layers, eliminating the need to train a separate inverse component for each layer.

## 5 Discussion

In this work, we set out to apply an efficient variant of the classic feature inversion approach from Dosovitskiy & Brox (2016) to study the intermediate representations of the TVMs DETR and ViT. We began by formulating a modular version of feature inversion that significantly improves efficiency by replacing large global inverse networks with lightweight, local inverse components, thereby substantially reducing the number of trainable parameters.

After a brief glimpse into image reconstructions obtained with our novel variant from DETR and ViT, we first validated its feasibility for network interpretability on both architectures. To this end, we qualitatively and quantitatively compared our approach to classical feature inversion on ViT and DETR, and variants of the two architectures fine-tuned with a combination of reconstruction loss and architecture specific objectives. We found that reconstructed images obtained with modular feature inversion are reflective of the processing mechanisms of both architectures. Additionally, despite potential error accumulation through repeated application of inverse components, the method opposes trivial image reconstructions, rendering our modular approach not only more efficient than classic feature inversion but also better suited for network interpretability.

Building on this foundation, we commenced with a systematic interpretability study of DETR and ViT, beginning with an investigation of how color information is processed. We observed that DETR progressively shifts object colors toward prototypical representations, while ViT preserves original color information throughout. Consistent with these findings, DETR shows strong robustness to color perturbations, whereas the classification performance of ViT degrades, challenging previous claims (Naseer et al., 2021; Paul & Chen, 2022) that ViT is remarkably resilient to image perturbations.

We continued our interpretability study with a focused analysis of how image structure is processed in the two architectures. We found that DETR abstracts object structure and context, modifying shapes and poses, omitting irrelevant but adding contextually relevant features, reflecting a shift toward prototypical representations that likely simplify object detection in later stages. In contrast, ViT retains object geometry and spatial layout with minimal distortion, pointing to a lower level of abstraction and a stronger preservation of visual detail.

We then turned towards analyzing spatial correlations between intermediate representations and input images. Using a novel analysis method in the context of feature inversion, specifically, injecting noise into intermediate representations, we found that ViT encodes spatial information in a localized manner. At the same time, DETR diffuses spatial information more globally. The spatial correspondence in ViT questions the importance of self-attention within the architecture, particularly given that we achieved reasonable classification accuracy in a ViT with disabled self-attention. Notably, Jaegle et al. (2021) have shown that a transformer-based model can achieve competitive accuracy on ImageNet-1k using only cross-attention. However, in their model, self-attention was still applied to register tokens, and its computational complexity exceeded that of ViT.

After briefly showing how DETR reconstructions vary with detection errors, we concluded our analysis by leveraging a key property of transformer architectures, namely, the constant shape of intermediate representations across encoder and decoder layers. The consistency allowed us to feed these representations into inverse components optimized for reconstruction from different layers. We found that both DETR and ViT

refine their representations gradually across layers, a pattern consistent with prior ViT studies (Raghu et al., 2021) and now extended to DETR, suggesting that gradual refinement is a general characteristic of TVMs. This property also enhances the efficiency of feature inversion in such models.

On the whole, we found both shared and divergent properties between DETR and ViT. For instance, both models exhibit gradually evolving representations across layers, yet differ in their treatment of color and structural image information. While similarities can likely be attributed to the transformer-based nature of both models, the differences raise important questions for future research, and we outline several hypotheses regarding their origin.

One potential factor is the difference in training data. The ViT we analyzed was trained on the large-scale JFT-300M dataset, which contains around 18,000 classes (Sun et al., 2017), while the DETR we analyzed was trained on COCO (Fleet et al., 2014), which includes approximately 90 object categories. The greater visual diversity in JFT-300M may require ViT to preserve finer image details, whereas DETR, trained on a smaller and more constrained set of categories, may afford more abstraction. Future work could examine how the training dataset influences feature inversion results, for example, by retraining both architectures on the same dataset.

Another hypothesis relates to the difference in training objectives. DETR is trained for object detection, which involves identifying and localizing multiple objects per image. In contrast, ViT is trained for image-level classification, where only a single object needs to be identified without localization. Classification, therefore, may place less demand on the model to transform input features extensively. In contrast, object detection requires more abstract and context-aware representations to support both recognition and spatial localization. Future research could explore this hypothesis by training the architectures with swapped objectives and analyzing the impact on feature inversion outcomes.

Lastly, architectural differences, particularly in the backbone, may also contribute to the observed differences in image reconstructions. DETR uses a CNN as its backbone, which is known to produce progressively more abstract representations (Mahendran & Vedaldi, 2014), whereas ViT employs an invertible linear embedding that preserves all input information. Consequently, the inputs to the transformer encoders in DETR and ViT differ substantially from the outset, potentially leading to distinct representational dynamics throughout the network. Future work could explore the role of the backbone by systematically swapping the backbones of the two architectures and analyzing their effect on image reconstructions.

From a methodological perspective, we have shown that modular feature inversion is both more efficient and more naturally aligned with the architecture of TVMs than the classical approach. These properties make it well-suited for analyzing modern iterations of TVMs such as DINOv2 (Oquab et al., 2024) or SAM (Kirillov et al., 2023). Furthermore, since our approach is not limited to TVMs and we expect it to offer advantages for a broad range of DNNs, it may also prove valuable for analyzing modern CNN-based models such as ConvNeXt V2 (Woo et al., 2023) or YOLOv8 (Ultralytics).

One particularly intriguing property of our method is that, despite yielding higher image reconstruction error than classical feature inversion, images are better suited for network interpretability. While we can attribute this effect to the closer mirroring of the forward processing path obtained with modular feature inversion compared to the classical approach, its extent remains unclear. Future research could explore this further by systematically varying the number of inverse components and examining the impact on both reconstruction quality and interpretability.

In this line of research, future work could also address a fundamental limitation of feature inversion: Even with the modular approach, it remains challenging to conclusively attribute specific properties in reconstructed images to individual processing stages. Drawing reliable conclusions typically requires additional quantitative analysis. However, increasing the number of inverse components may offer finer-grained insights and help localize specific representational effects to particular layers.

Our method may have broader applications beyond network interpretability. In the case of DETR, we observed that undetected objects often vanish in reconstructed images, while misclassified objects tend to appear significantly altered. These findings point to a promising direction for applying modular feature

inversion to error detection: By comparing reconstructed images to their inputs, discrepancies may serve as indicators of detection failures.

Drawing a parallel to computational neuroscience, prior work has shown that generative models of episodic memory require the integration of both discriminative and generative processes (Fayyaz et al., 2022). Future models could build on this idea by unifying a TVM and its inverse within a single architecture. Likewise, TVMs may be well-suited for biologically plausible learning systems, as they naturally support local reconstruction losses (Kappel et al., 2023).

In summary, we proposed a modular feature inversion framework for TVMs that enables scalable, component-wise interpretability with minimal training overhead. Applied to DETR and ViT, it revealed shared and distinct representational dynamics in abstraction, spatial encoding, and robustness. Beyond interpretability, the approach shows promise for error detection and biologically inspired learning, positioning modular inversion as a practical tool for probing modern vision models and guiding future discriminative-generative integration.

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

# A Disentangling architecture from objective

## A.1 Training DETR for Classification

In Section 4, we observed that DETR progressively abstracts object- and context-related features, shifting toward prototypical representations, whereas ViT preserves comparatively fine-grained, local details. To identify the source of these differences, we conducted an additional control experiment aligning the DETR training objective and training data with those of ViT. Specifically, we trained DETR for image classification on ImageNet-1K, keeping the architecture as intact as possible (i.e., preserving the backbone, encoder, and decoder) and computing the classification loss from a single object query.

The reconstructions shown in Figure 12 demonstrate that, despite the change in task and dataset, DETR continues to exhibit the same stage-wise trend toward abstraction and the formation of prototypical representations. These findings suggest that the observed processing differences between DETR and ViT are primarily attributable to architectural design, rather than to the detection objective or training data.

Beyond investigating the observed differences between DETR and ViT, we further analyzed prediction-head reconstructions from the DETR model trained for classification compared with the original DETR, to deepen our disentanglement analysis and highlight how detection and classification objectives shape representations differently at this stage within the same TVM architecture. At the prediction-head stage, reconstructions in the classifier DETR collapse to class-consistent prototypes. These reconstructions are the same across samples of the same class and distinct across classes, unlike the scene-consistent reconstructions we observed in the original DETR trained for detection. Moreover, when we feed the inverse prediction-head output back through the forward prediction head, the classifier DETR predicts the same classes. Together, these results indicate that the prediction-head reconstructions in the classification setting encode a shared, class-specific code independent of other details in the input scene.

These results contrast with our observations from the original DETR trained for the detection task, which retains richer input-dependent representations. Through our modular inversion analysis, we found that correlations and co-occurrences among objects are consistently preserved within the class and bounding-box channels across samples. Notably, this information is not limited to object-associated queries, as many queries labeled as "no object" (COCO background) also encode structured scene information, including cues about surrounding objects and context.

Our ablation studies (not presented in the original manuscript) on the bounding-box components of these "no object" queries, by zeroing positional coordinates, sizes (height/width), or both, show that modifying positional terms changes the reconstructed scene more than modifying sizes, indicating that position is the dominant carrier of scene structure in these representations. We will include a detailed analysis of these results in the appendix of the revised version.

Overall, these findings suggest that the detection task inherently reinforces scene-dependent representations, whereas aligning DETR to a global classification objective drives its prediction head toward a class-specific, scene-independent code.

In summary, our analyses reveal that DETR and ViT differ substantially in how they process inputs and form internal representations. These representational differences were previously unknown and, to our knowledge, have not been visually characterized in prior work. Our modular feature inversion framework provides an efficient direct visualization and stage-wise disentanglement of these inner mechanisms, offering a human-interpretable level of semantic and spatial insights that extends beyond existing interpretability methods, and can also inform them for more targeted and quantitative analyses by revealing both what and where representational changes occur. Through our experiments, we attributed the observed abstraction trends primarily to architectural design rather than task objectives. This understanding is not only relevant for guiding future architectural improvements but also carries practical implications for application domains where preserving detailed input information is critical, or where abstraction may be desirable for efficiency, and the model selection should therefore account for these representational tendencies.

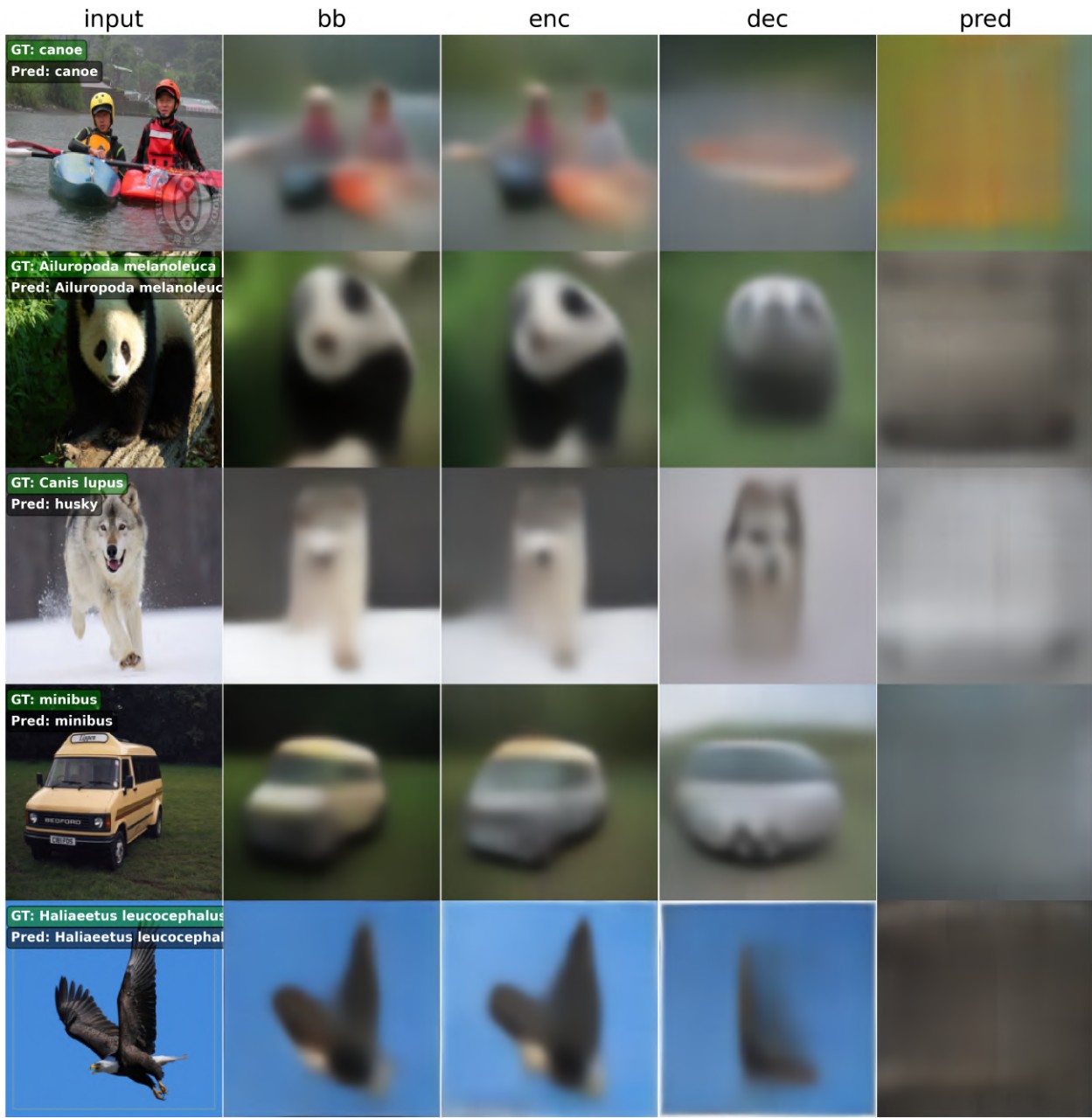

Figure 12: Reconstructions from different stages of DETR trained for classification.

## B Analysis on additional TVMs

We extended our experiments to the TVM DeiT III (Touvron et al., 2022), specifically the deit3_base_patch16_224 variant and SWIN Transformer (Liu et al., 2021), specifically the swin_base_patch4_window7_224 variant.

### B.1 DeiT III

Architecturally, DeiT III is equivalent to ViT, but its training procedure differs slightly. In particular, DeiT III employs a distinct data augmentation strategy, whereas ViT is trained on a much larger dataset

with only minimal augmentations. This augmentation strategy enables DeiT III to achieve comparable performance to ViT while being trained on significantly smaller datasets.

Consistent with our experiments on ViT in the main paper, we selected the same processing stages of interest (bb, enc) and used identical architectures for the inverse networks. Analogous to the experiments in the main paper, we present reconstructions for standard images (Figure 13), color-perturbed images (Figure 14), and manipulated images (Figure 15).

Across all reconstruction settings, DeiT III exhibits the same behavior as ViT. This consistency suggests that (a) our method is applicable to TVMs regardless of their specific training schemes, and (b) the detailed reconstructions observed when inverting ViT, compared to DETR, are not merely a consequence of the large-scale dataset used for ViT training, as we have hypothesized in the discussion.

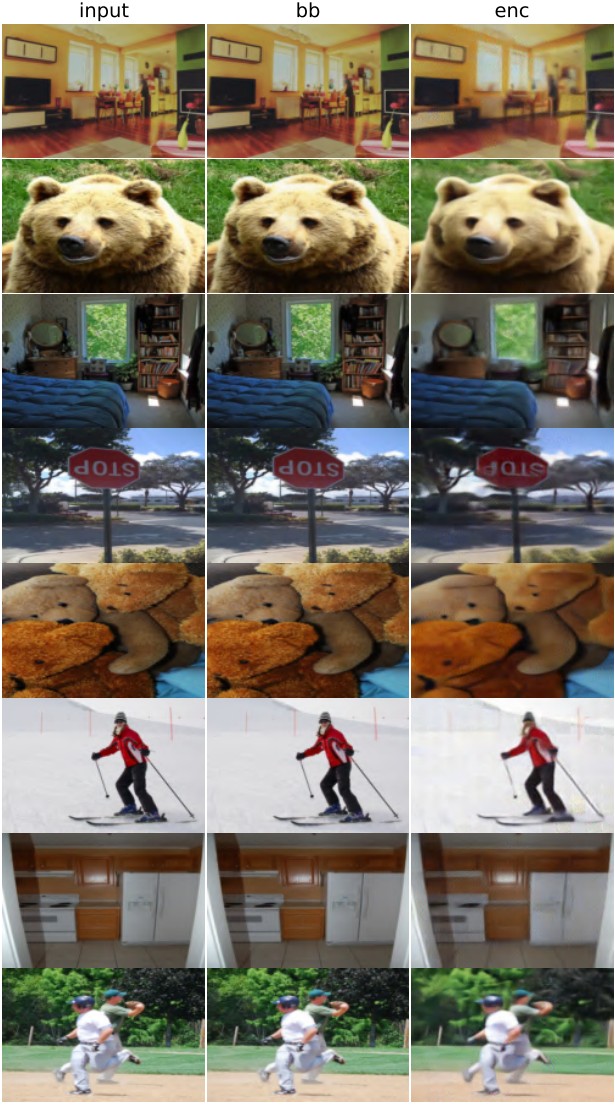

Figure 13: Image reconstructions from various processing stages of DeiT III.

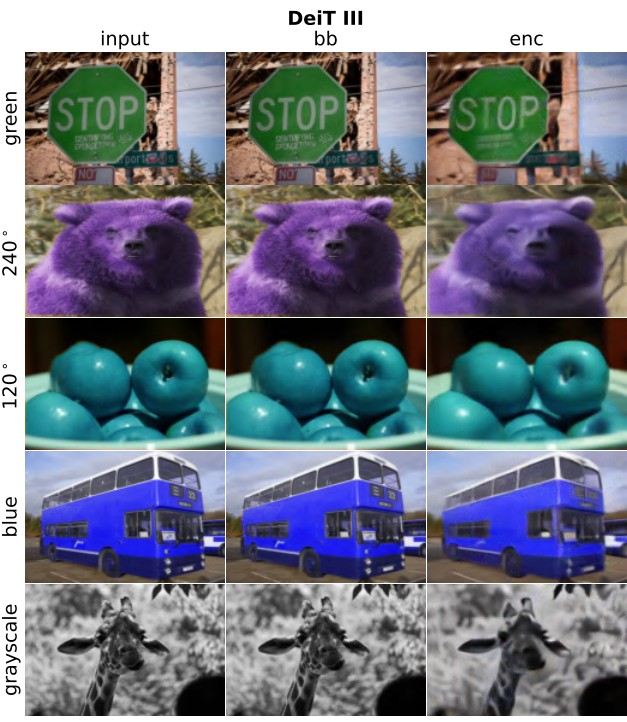

Figure 14: Image reconstructions from various processing stages of DeiT III on color-perturbed images, analogous to Figure 6.

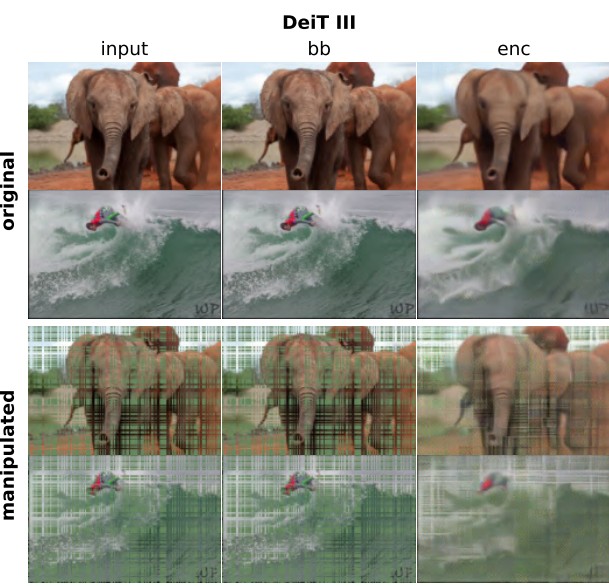

Figure 15: Image reconstructions from various processing stages of SWIN on randomly manipulated images, similar to Figure 9.

## B.2  SWIN

Swin Transformer (SWIN) (Liu et al., 2021) consists of a linear patch embedding (similar to that of ViT), which we refer to as bb for consistency, followed by five processing stages. Each stage is composed of a Swin Transformer Block. In contrast to the transformer blocks in ViT, self-attention in SWIN is computed

only locally within non-overlapping image windows, thereby reducing the computational complexity of the attention operation. Additionally, within each Swin Transformer Block, the windows are shifted and later shifted back, enabling a cross-window communication. Finally, the window size and patch size increase progressively across the processing stages.

For inverting SWIN, we selected the outputs of bb and the five processing stages as stages of interest. We used the same architecture for $bb^{-1}$ as for ViT. To invert the Swin Transformer Blocks, we employed architecturally equivalent blocks, but replaced the downsampling operations with upsampling operations, since, in the inverted information flow, the window and patch sizes decrease rather than increase. Analogous to the experiments in the main paper, we again present image reconstructions for standard images (Figure 16), color-perturbed images (Figure 17), and manipulated images (Figure 18).

Across all experiments, the image reconstructions are highly detailed across most processing stages, and the reconstructions under manipulations and color perturbations are qualitatively similar to those of ViT. Only the reconstructions from the last and second-to-last processing stages exhibit a loss of fine detail, slight color shifts, and distorted object outlines, with these effects being most pronounced in the final stage.

We interpret these results as follows. From our experiments in Section 4.5, we inferred that image details in ViT are propagated locally, suggesting an emphasis on locality in self-attention. This locality, which ViT must learn from large-scale data, is already inductively encoded in SWIN through its local windowing mechanism. Consequently, it is unsurprising that SWIN behaves similarly to ViT and can be trained on smaller datasets, as the model's architecture inherently enforces the locality that ViT must acquire through data. The loss of image detail in the final layer is likely due to the average pooling operation applied to the output of stage five prior to the classification head.

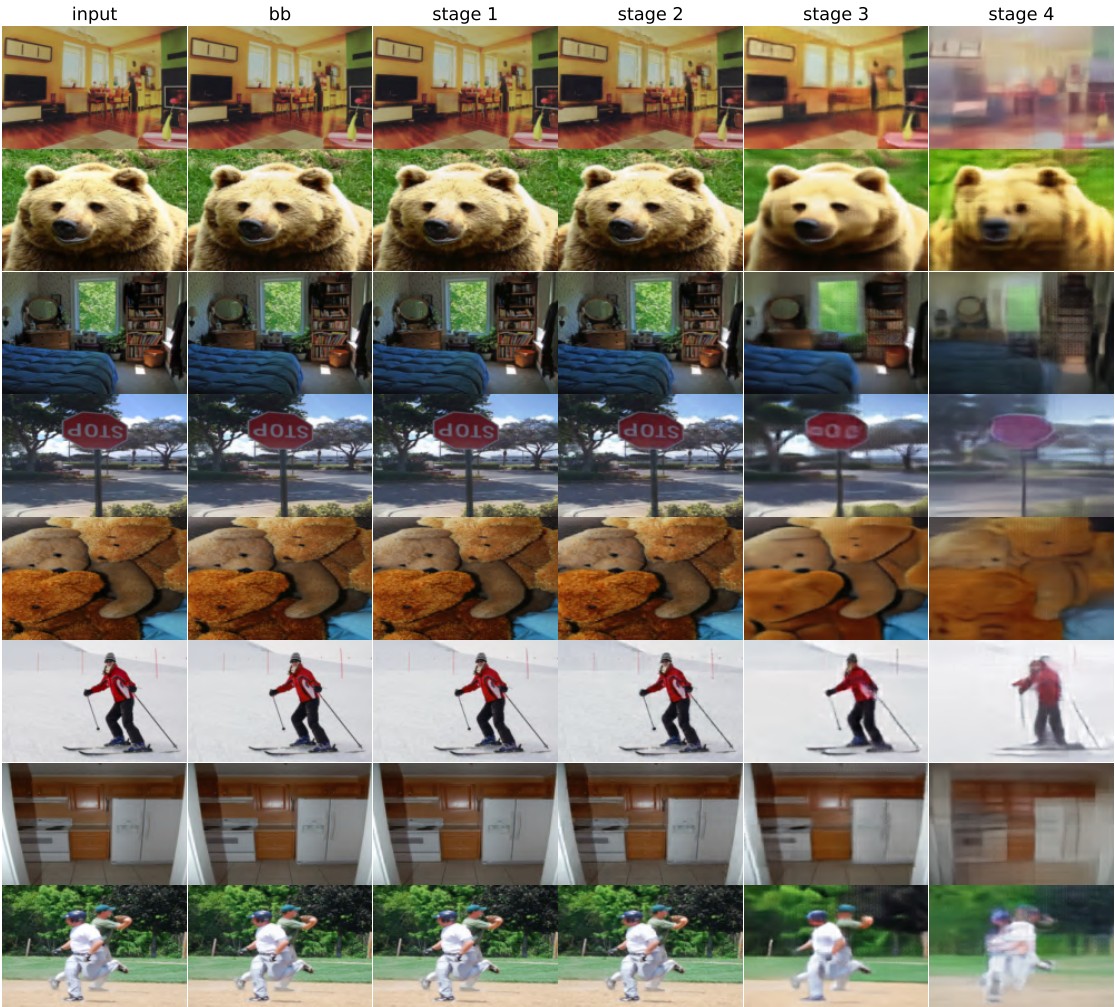

Figure 16: Image reconstructions from various processing stages of SWIN.

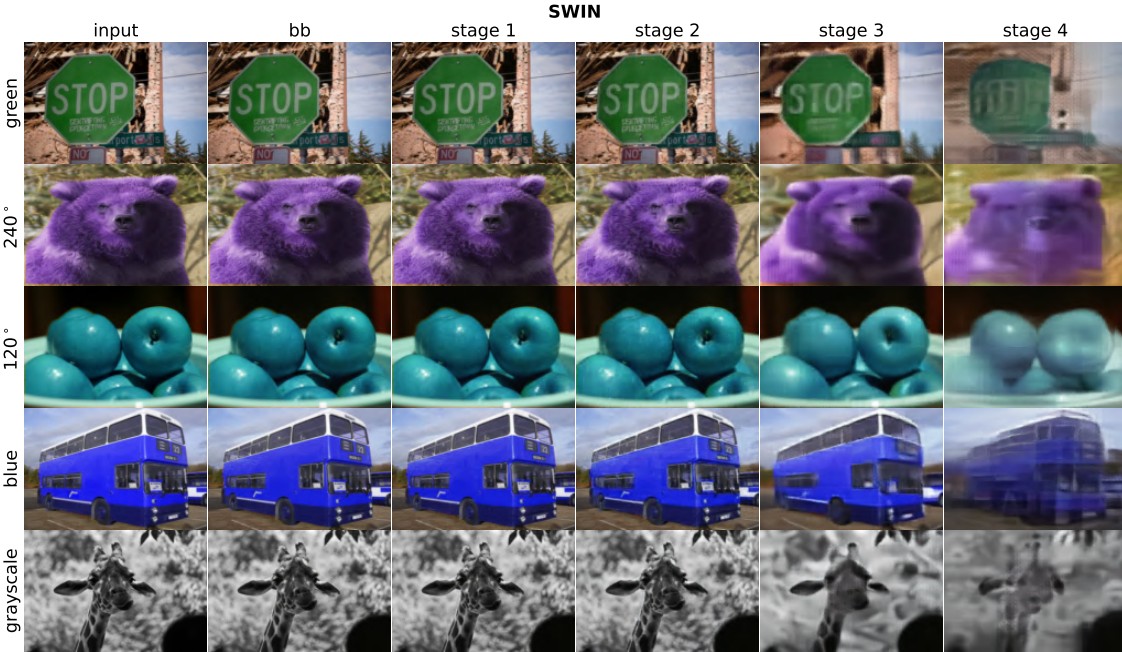

Figure 17: Image reconstructions from various processing stages of SWIN on color-perturbed images, analogous to Figure 6.

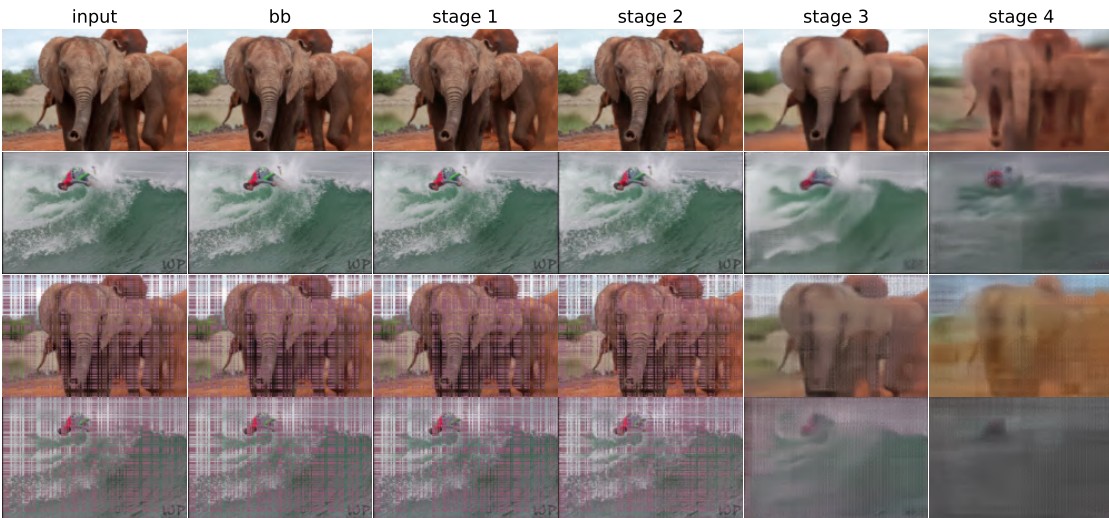

Figure 18: Image reconstructions from various processing stages of SWIN on randomly manipulated images, similar to Figure 9.

