# OpenReview forum: "Understanding Transformer-based Vision Models through Inversion"
_TMLR — Rejected by TMLR_

### Review · Reviewer_RzjC · 2025-09-05

**Summary Of Contributions:**

**[Summary]**
The paper introduces modular feature inversion, training small, local inverse components and chaining them, to interpret transformer-based vision models efficiently. Applied to DETR and ViT, the method reconstructs images from multiple stages and reveals differences: DETR increasingly abstracts structure and pushes colors toward prototypical hues at higher layers, while ViT preserves fine detail and original colors throughout. Quantitative tests (e.g., color perturbations) show DETR is robust to color shifts whereas ViT’s accuracy drops; there’s also a documented trade-off where fine-tuning for better reconstructions can hurt task performance. Overall, the approach scales more efficiently than classic inversion and supports stage-wise diagnosis and analysis.

**[Strengths]**
- A clear insight shows DETR's color shift and structural abstraction.
- Comprehensive analysis over different aspects of DETR and ViT.
- The stage-wise design can help trace the detection errors of DETR.

**[Weaknesses]**
- The inverse network analyses are largely descriptive and it is unclear how they translate into practical design choices. The reviewer believe this is another major drawback of the inverse network. The prior inversion work rarely informed concrete architecture or training changes in modern models (ResNets, Transformers), and the same risk applies here: what should a researcher do differently given the observed ViT–DETR differences? The paper would be stronger with examples that turn insights into decisions (e.g., loss design, data augmentation, curriculum, pruning/debug rules).

- The "fundamental" contrast between ViT and DETR could simply result from task and architecture differences (classification vs. detection supervision, decoder effects) rather than an some inherent representational gap. Without a better controlled comparisons, matched pretraining data/augments, with/without decoder, frozen backbones, ViT fine-tuned for detection vs. DETR-style training, the conclusion risks being trivial (“detection just needs higher-level cues”).

- The paper argues efficiency over prior methods as it adapts layer-wise inversion by concatenating multiple layers and training end-to-end. However, similar concatenation with shared-optimizer through all components is potentially also applied to the baseline as well. This makes the advantage look just in implementation rather than fundamental modification. Provide controlled comparisons and report such as memory, FLOPs, and parameter counts to substantiate the efficiency claim.

**Additional Comments:**

The additional experiments I’m requesting are primarily about efficiency. For the ViT–DETR comparison, the aim is diagnostic, i.e. control for training strategy (objectives, decoder, data/augments) to identify the root causes so the findings are more broadly applicable.

My main concern (Weakness 1) is limited practical usage: the inverse network currently delivers compelling visualizations, but these remain largely qualitative and don’t yet translate into concrete training or design decisions. Strengthening the bridge from observation to action (e.g., loss/augmentation tweaks) would substantially increase the paper’s utility.

I suspect this gap, more than efficiency alone, explains why prior inversion work has seen limited downstream adoption. I wish a further discussion with the author on this issue if possible.

**Audience:**

Yes

**Audience Explanation:**

Some TMLR’s audience would care, especially for those who works in interpretability/analysis, vision transformers (ViT/DETR), and robustness/diagnostics. The modular inversion idea is methodologically interesting on its own and the ViT–DETR contrasts are the kind of empirical insight many researchers like to see.

**Broader Impact Concerns:**

The paper targets to analyze the existing methods. THere is no Broder Impact Statement needed.

**Claims And Evidence:**

No

**Claims Explanation:**

The author claimed 4 major contributions in the introduction.

-  The paper claims improved efficiency, but provides no empirical validation. Please add controlled measurements, memory peak, FLOPs, and parameter counts, plus ablations on grouping depth. The supplement appears to include code only; quantitative results are required to substantiate this claim

- For the second, the manuscript convincingly demonstrates how reconstructed images can be used to interrogate internal processing, with clear stage-wise analyses. This is a substantive contribution to interpretability.

- For the third, the reviewer acknowledges the reported similarities. To strengthen the claim, consider quantifying how consistently these properties hold across variants (e.g., different backbones, data regimes).

- For the last, the asserted fundamental differences likely due to training strategy and architecture (e.g., detection losses, decoder effects). It requires more comparisons, matched pretraining, with/without decoder, frozen vs. fine-tuned backbones, to localise the root cause of it.

**Requested Changes:**

1. A training and inference cost comparison is needed as author claimed the efficiency of the proposed method

2. Controlled ViT–DETR comparisons to disentangle training vs. architecture. For example, using the same pretraining data/augmentations or use the same architecture but training in different loss.

3. If possible, add another transformer-based model, such as swin.

---

> ### Author Response · Authors · 2025-10-17
> **Nature of Analysis and Control Experiment**
>
> We appreciate the reviewer’s thoughtful feedback and the time dedicated to evaluating our work. Our responses are provided below.
>
> **Descriptive nature of analyses:** Our analyses indeed focus on understanding how DETR and ViT behave. While not our priority, we believe our method has the potential for enhancing network architectures. We illustrate this potential by comparing the insights from our analysis with enhancements of ViT in the literature. For example, the low level of abstraction observed in ViT, i.e., a propagation of overly detailed image information, suggests that ViT could be made more efficient by biasing the model toward higher-level representations, as done in the PVT [1]. Similarly, the predominantly local processing observed in ViT indicates that self-attention could be applied more efficiently locally, as demonstrated in SWIN [2]. Finally, the sensitivity of ViT to color perturbations suggests that it could benefit from stronger image augmentations during training, as shown by [3].
>
> **Controlled ViT–DETR comparisons:** Although this was not our initial goal, we agree that, given the observed differences between DETR and ViT, a control experiment disentangling the effects of architecture and objective is of interest. We will provide the corresponding results as soon as possible. For a detailed discussion of this point, please refer to the paragraph "Choice of Vision Models" in our response to reviewer bxVc, who raised a similar request.
>
> **References**
>
> [1] Wang et al. (2021) Pyramid Vision Transformer: A Versatile Backbone for Dense Prediction without Convolutions
>
> [2] Liu et al. (2021) Swin Transformer: Hierarchical Vision Transformer using Shifted Windows
>
> [3] Touvron et al. (2020) Training data-efficient image transformers & distillation through attention

---

> > ### Comment · Reviewer_RzjC · 2025-10-19
> >
> > Dear Authors, thanks to your reply. The following is my response.
> >
> > **Response to Descriptive nature of analyses**: I share the same concern as reviewer FtAT. Neither of us argues that pursuing understanding is wrong; rather, we question whether the level of understanding presented in your work is sufficiently deep to inspire meaningful future designs. If one of the main purposes of this paper is to provide understanding, then that understanding should be insightful and inspirational.
> >
> > However, several of the papers you cited were published before your analyses, yet they already proposed improvements to ViT or DETR based on intuition or alternative reasoning. This, in fact, reinforces our concern: if those advancements could occur without your proposed understanding, what unique or indispensable value does your analysis add? In other words, what distinguishes your “understanding” from prior understanding? What can your analysis enable that previous intuitions or heuristics could not?
> >
> > **Response to Controlled ViT–DETR comparison**: I have read your response to reviewer bxVc, but I do not agree that performing controlled experiments across different tasks or architectures should be considered merely as addressing a reviewer’s/reader's curiosity or deferred to future work. If one of your main contributions lies in understanding (unless I misunderstood and your work is primarily about making an existing method more efficient), it is crucial to clarify the scope and generality of that understanding.
> >
> > Specifically, is your analysis task-specific? Does it depend on architectural details? Without such clarification, the value of your claimed understanding remains incomplete. These experiments are not optional; they are essential to substantiate the contribution of your paper. I look forward to seeing additional results or discussion to strengthen this part you will provide later.

---

> ### Author Response · Authors · 2025-10-17
> **Efficiency**
>
> **Efficiency:** We thank the reviewer for requesting a more detailed quantitative analysis of the efficiency of our method. Below, we provide an extended comparison focusing on DETR, since we analyzed four processing stages in this architecture versus two in ViT. Consequently, our method yields more pronounced benefits on DETR.
>
> The DETR variant we analyzed, DETR-R50, consists of approximately $\approx 4.15 \times 10^6$ parameters in total (backbone: $\approx 2.35 \times 10^6$, transformer encoder: $\approx 0.79 \times 10^6$, transformer decoder: $\approx 0.95 \times 10^6$ prediction head: $\approx 0.02 \times 10^6$). As noted at the bottom of Section 3.2, following the practice introduced in feature inversion by Dosovitskiy and Brox [*Inverting Visual Representations with Convolutional Networks, 2016*], inverse networks should approximately mirror the structure of their respective forward networks.
>
> In our modular approach, we adopted this principle, with one exception. While the inverse encoder, decoder, and prediction head mirror their forward counterparts, we used a larger inverse backbone, which substantially improved reconstruction performance. This effect was not observed for other stages. Specifically, our inverse backbone comprises $\approx 7.87 \times 10^6$ parameters, while the inverse encoder, decoder, and prediction head comprise $\approx 0.79 \times 10^6$, $\approx 0.95 \times 10^6$, and $\approx 0.02 \times 10^6$ parameters, respectively. Thus, inverting DETR using our modular approach from all four stages of interest requires a total of $\approx (7.87 + 0.79 + 0.95 + 0.02) \times 10^6 = 9.63 \times 10^6$ parameters.
>
> In the classical approach, adhering to the convention that each inverse model mirrors its corresponding forward path, the parameter count grows cumulatively. Specifically, inversion from the backbone stage involves $\approx 7.87 \times 10^6$ parameters. Inversion from the encoder stage corresponds to a concatenation of an inverse backbone and inverse encoder, i.e., $\approx (7.87 + 0.79) \times 10^6$. Following this logic, the total parameter count for the classical approach is: $(4 * 7.87 + 3 * 0.79 + 2 * 0.95 + 1 * 0.02) \times 10^6 = 35.77 \times 10^6$. Hence, our method requires only $\approx 27$ \% of the parameters of the classical method—consistent with our theoretical analysis in Section 3.2. We note that the parameter count for both approaches could be reduced through careful tuning; thus, the reported values represent empirical observations rather than strict lower bounds. The larger inverse backbone largely accounts for the difference, as it is included in each inverse model in the classical approach. For other architectures, the gap may be smaller.
>
> For comparing training speed, we trained inverse networks for both approaches using the same pipeline in accordance with standard deep learning practices. All experiments were conducted on an NVIDIA A100 GPU with the COCO 2017 dataset. Input images were resized to 640 x 480, and the batch size was set to 32, constrained by memory requirements. We used the Adam optimizer with default parameters, tuning only the learning rate. A practical and efficient training strategy for both methods is to compute the intermediate embeddings at the processing stages of interest using DETR, reconstruct these embeddings sequentially with the respective inverse models, compute the reconstruction losses, and update all parameters jointly. With this setup, one training epoch (i.e., one pass through the dataset) required 119 minutes for the classical approach and 57 minutes for our approach. Thus, image throughput during training is roughly twice as high for our method.
>
> Comparing convergence speed between the two methods is less straightforward. With larger networks, the classical approach tends to overfit more quickly, e.g., after about 18 epochs for reconstructions from the decoder stage and 20 epochs for reconstructions after the encoder stage. In contrast, the smaller modular networks continued to show modest improvements even after 50 epochs. However, before overfitting, the classical approach still produced reconstructions of poor quality, both visually and in terms of MSE, as reflected by the complete breakdown of DETR’s object detection performance on the reconstructed images. We were able to improve the reconstruction quality ($\approx$ 10\% in terms of MSE) in the classical approach by initializing its inverse networks with those pretrained using our modular approach, and then fine-tuning them, as described in Section 4.2 of the main paper. However, this substantially increased total training time, since it requires first training the modular inverses and then fine-tuning them.
>
> Regarding inference speed, both methods perform comparably. While image throughput is slightly higher for the modular approach, both can reconstruct images within a few seconds on standard consumer hardware.

---

> > ### Author Response · Authors · 2025-10-17
> > **Limited application**
> >
> > **Limited application:** We believe that feature inversion has not been widely applied for two main reasons. Firstly, efficiency is a significant limitation, particularly when researchers are interested in multiple processing stages, as this would require training large networks for each stage. Based on both the theoretical arguments presented in the paper and the new quantitative evidence provided in the paragraph above, we can confidently say that our approach offers a substantial improvement in this regard. Secondly, images obtained with the classical approach are often difficult to interpret. This challenge is also evident in the original feature inversion paper by Dosovitskiy and Brox [*Inverting Visual Representations with Convolutional Networks, 2016*], where the qualitative insights of the method are limited, and the authors primarily demonstrate that networks can, in principle, be inverted using classical feature inversion. As we have shown qualitatively in the paper and now quantitatively in the paragraph "MSE as Criterion" in our response to reviewer bxVc, our method produces semantically richer images that facilitate model interpretability and may ultimately improve architectures. We hope that these two advancements will encourage broader adoption of the method in the future.

---

> > ### Comment · Reviewer_RzjC · 2025-10-19
> >
> > Thanks for the replay and understood. Then my concerns to the efficiency could be considered as addressed.

---

> ### Author Response · Authors · 2025-10-28
> **Update on requested experiments**
>
> We thank the reviewer for detailed comments and constructive feedback. We have conducted additional experiments addressing the raised points (For ease of review, these preliminary (unpolished) results are included in the Appendix of the revised manuscript):
>
> **1. Disentangling architecture from objective in DETR–ViT comparisons**
> As promised, we trained a classifier variant of DETR on ImageNet-1K instead of COCO to examine whether DETR’s prototypical abstraction persists without the detection objective. For details, please refer to Section A in the appendix.
>
> **2. Generality across TVMs**
> To assess the generality of our modular feature inversion method, we applied it to DeiT-III [1] and SWIN [2], analyzing whether ViT-like detail preservation extends beyond original ViT. Also, the local image details observed in ViT provide further insight into why SWIN achieves strong performance with local self-attention alone. For details, please refer to Section B in the appendix.
>
> **3- Value of our method**
>
> A unique property of our method (and of feature inversion in general) is its focus on analysing the preservation, or abstraction, of image details, thereby offering additional insight into processing mechanisms that have not been previously considered. For example, the local image details observed in ViT provide further insight into why SWIN achieves strong performance with local self-attention alone. Additionally, analysis of image details can help guide the selection of vision model backbones for downstream tasks, depending on whether the downstream task benefits more from detailed visual information or from abstract representations.
>
> **References**
>
> [1] Touvron, Hugo, Matthieu Cord, and Hervé Jégou. "Deit iii: Revenge of the vit." European conference on computer vision. Cham: Springer Nature Switzerland, 2022.
>
> [2] Liu, Ze, et al. "Swin transformer: Hierarchical vision transformer using shifted windows." Proceedings of the IEEE/CVF international conference on computer vision. 2021.

---

### Review · Reviewer_FtAT · 2025-09-27

**Summary Of Contributions:**

This paper proposes a novel modular feature inversion framework designed to interpret the inner workings of transformer-based vision models, particularly DETR and ViT . The key contributions of the paper include:

1. Introduction of Modular Feature Inversion:

   Instead of training large inverse networks for each layer as in previous works, the authors propose a modular approach that trains local inverse components. This design improves computational efficiency, scalability, and interpretability.

2. Systematic Application to DETR and ViT:

   The proposed method is empirically validated on DETR and ViT, with reconstructed images from different stages used for both qualitative and quantitative analysis.

3. New Analysis Techniques:

   The authors introduce methods like targeted embedding manipulation, color perturbation analysis, and spatial token ablation to interpret intermediate representations in TVMs.

4. Key Insights:

   - DETR abstracts image features into prototypical representations at deeper layers, losing fine details but gaining robustness.
   - ViT retains fine-grained visual information across layers, maintaining spatial correspondence.
   - DETR shows greater robustness to color perturbations, while ViT is more sensitive.
   - ViT exhibits spatial locality, while DETR encodes information more globally.

**Strengths**

- The paper presents a modular feature inversion framework and applies it systematically to two representative transformer-based vision models, ViT and DETR. The proposed design aligns well with the architectures of these models and demonstrates practical applicability.
- The authors conduct multi-faceted analyses focused on color processing, structural abstraction, and spatial correspondence. Both qualitative and quantitative results are used to support the interpretation of intermediate representations.
- The experimental design is thoughtful and provides insights into how ViT and DETR differ in their visual information processing—for example, DETR’s abstraction of color and detail versus ViT’s preservation of spatial locality.
- The paper is clearly written and well-structured, with visualizations that effectively support the presentation of results and make the analyses accessible to the reader.

**Weaknesses**

- While the modular inversion framework is well-implemented, the method itself is relatively straightforward and primarily reorganizes existing ideas, without introducing fundamentally new modeling mechanisms or algorithmic components.
- The experimental evaluation is limited to ViT and DETR, without validation on more diverse or larger-scale transformer models such as SAM or DINOv2, which limits the generality of the conclusions.
- The interpretability analysis relies heavily on visual inspection of reconstructed images, with limited use of systematic quantitative metrics, which may impact the reproducibility and objectivity of the findings.
- Although the modular design improves training efficiency, the current framework still makes it difficult to assign specific semantic or visual phenomena to individual model layers, resulting in relatively coarse-grained explanations.

**Additional Comments:**

The paper is clearly written and well-structured. The proposed modular feature inversion framework is thoughtfully designed and adapted to the architecture of transformer-based vision models. The visualization results are effective in conveying the differences in representational behavior between ViT and DETR, and the analysis covers a diverse set of interpretability angles.

That said, the scope of the evaluation is relatively narrow. Extending the method to additional architectures would enhance its generality. Furthermore, relying mainly on visual inspection for interpretability limits the objectivity of the findings. Introducing standardized quantitative measures for reconstruction quality would make the analysis more robust.

Overall, the work offers a useful framework and valuable insights into transformer representations. With additional experiments and more systematic evaluation metrics, it could serve as a strong reference for future research in model analysis and interpretability.

**Audience:**

Yes

**Audience Explanation:**

The findings of this paper would be of interest to TMLR’s audience because transformer-based vision models such as ViT and DETR are widely used in both academic research and practical applications. Understanding their internal representations and processing mechanisms is a key topic for the machine learning community, particularly for researchers working on model interpretability, robustness, and architecture design. The paper provides systematic analyses of color processing, structural abstraction, and spatial correspondence in these models, which offer valuable insights that go beyond performance metrics and help inform future work on vision transformers.

**Broader Impact Concerns:**

I do not have major concerns regarding the broader societal or ethical implications of this work. The paper focuses on interpretability and analysis of existing vision transformer models through feature inversion techniques, which are primarily diagnostic in nature. The method does not introduce any new data collection practices or deployment frameworks that would raise immediate ethical concerns.

**Claims And Evidence:**

Yes

**Claims Explanation:**

The claims made in the submission are supported by clear and consistent evidence. The authors provide both qualitative and quantitative analyses to demonstrate the effectiveness of the proposed modular feature inversion framework when applied to ViT and DETR. Visual reconstructions across model layers, reconstruction error measurements, color perturbation tests, and token manipulation experiments are all appropriately designed to support the paper’s central conclusions. Although the evaluation is limited to two models and the interpretability analysis relies heavily on visual inspection, the evidence presented is sufficient to validate the claims within the intended scope of the work.

**Requested Changes:**

1. **Include additional transformer-based vision models in the evaluation**

   This change is critical for acceptance. The current experiments are limited to ViT and DETR. While these models are representative, the paper would benefit significantly from evaluating the proposed method on a broader range of transformer-based vision models, such as DINOv2 or SAM. This would provide stronger support for the general applicability and robustness of the framework.

2. **Add quantitative metrics to assess reconstruction quality**

   This change is recommended to strengthen the work. The interpretability analysis relies primarily on qualitative visualizations. Including standard quantitative metrics such as SSIM (Structural Similarity Index) and PSNR (Peak Signal-to-Noise Ratio) would make the evaluation more rigorous and help readers objectively compare reconstruction performance. While not strictly necessary for acceptance, it would improve the clarity and reproducibility of the results.

---

> ### Author Response · Authors · 2025-10-17
> **Response to Reviewer Reviewer FtAT**
>
> We thank the reviewer for their detailed review and suggestions to improve the paper. We greatly appreciate the time and effort put into reviewing our work. Below, we address the points raised.
>
> **Limitation to ViT and DETR:** We agree that demonstrating the applicability of our method across a broader range of models generally strengthens claims about its generalizability. However, as Reviewer bxVc also noted, DETR and ViT are architecturally quite distinct, which already provides substantial evidence for the generalizability of our modular feature inversion approach.
>
> While extending our method to more recent TVMs such as DINOv2 or SAM would certainly be interesting, their base models are not substantially larger than ViT or DETR in terms of parameter count (ViT-B: $86 \times 10^6$, DETR-R50: $42 \times 10^6$, SAM-ViT-B: $100 \times 10^6$, SwinV2-B: $88 \times 10^6$). Consequently, such experiments would not necessarily provide stronger evidence for scalability to larger models.
>
> That said, applying our approach to larger variants of modern TVMs would be valuable. However, conducting such an analysis is not feasible within the four-week rebuttal period. Instead, we will include an additional analysis of a DETR model trained for classification, thereby disentangling architecture from objective (please see our response to Reviewer bxVc, "Choice of vision models" paragraph). We are currently finalizing training and will provide results as soon as possible below this comment.
>
> **Objectivity of findings:** Thank you for suggesting including additional metrics to strengthen the objectivity of our findings. We have expanded our analysis accordingly and provide details in the paragraph “MSE as a Criterion” in our response to reviewer bxVc. The new metrics quantify a phenomenon we previously described only qualitatively: Notably, while our method yields a higher MSE than classical feature inversion, it produces semantically richer images, consistent with our qualitative assessment.

---

> ### Author Response · Authors · 2025-10-28
> **Update on requested experiments**
>
> We have conducted additional experiments addressing the raised points (For ease of review, these preliminary (unpolished) results are included in the Appendix of the revised manuscript):
>
> **1. Disentangling architecture from objective in DETR–ViT comparisons**
> As promised, we trained a classifier variant of DETR on ImageNet-1K instead of COCO to examine whether DETR’s prototypical abstraction persists without the detection objective. For details, please refer to Section A in the appendix.
>
> **2. Generality across TVMs**
> To assess the generality of our modular feature inversion method, we applied it to DeiT-III [1] and SWIN [2], analyzing whether ViT-like detail preservation extends beyond original ViT. Also, the local image details observed in ViT provide further insight into why SWIN achieves strong performance with local self-attention alone. For details, please refer to Section B in the appendix.
>
> **References**
>
> [1] Touvron, Hugo, Matthieu Cord, and Hervé Jégou. "Deit iii: Revenge of the vit." European conference on computer vision. Cham: Springer Nature Switzerland, 2022.
>
> [2] Liu, Ze, et al. "Swin transformer: Hierarchical vision transformer using shifted windows." Proceedings of the IEEE/CVF international conference on computer vision. 2021.

---

### Review · Reviewer_bxCv · 2025-10-05

**Summary Of Contributions:**

This paper introduces a novel feature inversion method to reconstruct images. Instead of the classical approach of feature inversion where every inverse component is trained to reconstruct an image, the authors train smaller inverse components that invert a single processing stage which are then chained together to reconstruct the full image.

The authors use their new approach on two transformer based vision models — Detection Transformer (DETR) and Vision Transformer (ViT). The authors generate reconstructed images for these two models at different processing stages. For DETR: encoder backbone (bb), encoder (enc), decoder (dec), prediction (pred)  and for ViT: backbone (bb), encoder (enc).

They systematically analyze the two models across different properties — color, structure, spatial correlations, detection errors, and intermediate layers.

The authors found these main results from their empirical experiments:

- DETR progressively abstracts visual information in later layers while ViT retains fine-grained detail throughout the layers.
- DETR shows robustness to color perturbations whereas the classification performance of ViT degrades under similar conditions.
- DETR diffuses spatial information globally across tokens, while ViT encodes it in a more localized manner.

The authors position this framework as valuable tool for interpretability and for probing modern vision models.

**Strengths:**

- The paper introduces a novel way of efficiently performing feature inversion by training smaller local inverse components which are then chained together to produce the reconstructed image.
- The authors conduct a wide range of useful empirical experiments and study different properties of images on two vision models through their framework
- The paper introduces an interesting way of probing modern vision models with potential applications in error probing and interpretability.

**Weaknesses:**

- **Choice of vision models:**  DETR and ViT have fairly different model architectures making the comparison between them difficult. So the results obtained in this paper are less compelling as the amount of impact of architectural changes is unclear. The authors could have grounded the  experiments better by (a) using models with similar architectures (ex- two ViT variants) but different tasks (detection vs. classification), or (b) using different architectures while keeping the training task constant. This would have enabled a more controlled and insightful comparison.
- **MSE as a criteria:**  MSE is a pretty weak criteria to judge the similarity between the reconstructed image and the input image; as the authors themselves note in the paper, high MSE scores do not always correlate with poor perceptual quality leading to ambiguity that must be resolved with qualitative descriptions. This confusion in the paper could have been handled quantitively by using better metrics to compare the final reconstructed image against the original image. I can understand the idea of using MSE to train the actual inverse components, but the final reconstructed image could be compared against the original image using similarity measures like — Mean Absolute Error (MAE), Structural Similarity Index (SSIM), Learned Perceptual Image Patch Similarity (LPIPS). This would provide a more robust quantitative foundation for the paper's claims, reducing the reliance on subjective visual assessment and reduce the confusion of having contradicting results “high MSE but it is still visually similar”.
- **Corroborating evidence found through feature inversion using other interpretability techniques:**  To give a more compelling evidence, the authors could have used other techniques like activation maximization in support of the results obtained through their feature inversion method. For example, say we get the reconstructed image from enc through inversion. Then, activation maximization could have been done at the layers of encoder to get additional visualizations. By using both these evidences more interesting conclusions could have been made about the results.
- **Improvements in computational efficiency:** The paper claims that the modular inversion method offers significant improvements in computational efficiency, but this is not substantiated with quantitative evidence. The authors could have included metrics such as the total number of trainable parameters for the inverse models (modular vs. classic), comparative training times, or FLOPs to empirically validate the proposed efficiency benefits.

**Audience:**

Yes

**Audience Explanation:**

The paper introduces a new, novel way of performing feature inversion to interpret modern vision models and probe them. This technique is interesting and many in TMLR's audience would find it interesting.

**Claims And Evidence:**

Yes

**Claims Explanation:**

The claims in the paper are supported by the extensive empirical experiments conducted and the ample explanation of each experiment.

**Requested Changes:**

It would be great to see quantitative measures like MAE/SSIM/LPIPS used to get better statistics on the reconstructed images to better support the visual analysis and conclusions.

If possible, also add brief discussions in Appendix for the remaining points mentioned in the Weaknesses section.

In Fig 4, the center and the right plots should have the heading or labels specifying “DETR” and “ViT” to make it clearer, otherwise the plots are difficult to interpret without referring back to the caption.

---

> ### Author Response · Authors · 2025-10-17
> **Choice of vision models**
>
> We thank the reviewer for their thorough assessment and valuable suggestions to improve our paper. We sincerely appreciate the time and effort dedicated to reviewing our work. Below, we provide detailed responses to the points raised.
>
> **Choice of vision models:** In interpretability research, studies typically address three complementary scopes: (1) understanding how a model behaves, (2) investigating why that behavior arises, (3) improving the behavior toward desired outcomes. Our work primarily addresses (1) by introducing a modular feature-inversion framework for systematic analysis of transformer-based vision models, which visually characterizes how representations evolve across stages. It also contributes to (2) by providing stage-wise views of internal representations that localize where changes occur and attribute behavior to specific processing stages.
>
> We selected DETR and ViT as representative architectures to demonstrate the general applicability of our approach, as they differ substantially in both design and task: DETR is an encoder–decoder model trained for object detection, whereas ViT is an encoder-only model trained for image classification. Our initial goal was not to conduct a direct comparison but rather to test whether the proposed feature inversion technique could provide meaningful insights across such diverse transformer-based vision models. We did not anticipate, however, that their differences would become so clearly evident through the inverted representations.
>
> Given the observed differences using our technique, we agree that a systematic comparison focusing on architectural, task-specific, and training factors would provide valuable additional insights. As noted in our discussion, we had originally planned to pursue this in future work. Nevertheless, given your and reviewer RzjC’s interest, we have conducted an additional control experiment in which DETR is trained for classification, illustrating how our technique can support further investigations of architectural, task-related, and training factors. Since this experiment requires additional time to fully converge and produce stable results, we are currently finalizing the training and will provide the corresponding results as soon as possible.

---

> ### Author Response · Authors · 2025-10-17
> **MSE as criterion**
>
> **MSE as criterion.** We thank the reviewer for pointing out the limitation of exclusively relying on MSE for evaluation, and the narrow perspective it provides. We agree that stronger and more perceptually aligned measures can provide clearer quantitative support for our claims. Therefore, we have extended our analysis to include other common metrics such as SSIM, PSNR, LPIPS, FID, and CLIPScore.
>
> SSIM and PSNR primarily assess structural and pixel-wise similarity, reflecting how closely two images match in local detail and appearance. LPIPS, though more perceptually grounded, still applies weighted averaging of features across hierarchical layers of a pretrained network, making it partly sensitive to low-level texture similarities. In contrast, FID and CLIPScore only operate on deep semantic features. FID measures distributional distances in latent feature space, while CLIPScore compares deep embeddings from a model pretrained to align images with textual descriptions, which captures object-level consistency. Additionally, we evaluate DETR Average Precision (AP) by feeding reconstructed images back into DETR and computing AP on the reconstructions for both the classic inversion and our modular approach. This object-level metric assesses how well each method preserves information needed for detection, complementing other measures. While our method does not achieve pixel-level precision, as reflected by metrics more sensitive to fine-grained structure, such as MSE, SSIM, and LPIPS, it performs significantly better on perceptual metrics, including FID, CLIPScore, and AP compared to classic inversion. These quantitative results support our claim that the proposed method yields images that retain stage-specific transformations, are semantically richer and easier to interpret, and do not collapse into the globally averaged outputs produced by classical feature inversion. We will include these comprehensive additional metrics (see the table below), and an expanded discussion in the revised manuscript.
>
> **Table: Quantitative comparison of reconstruction quality across modular and end-to-end models on the COCO validation set.**
> Mean image baseline denotes the average reconstruction error between validation images and the mean image of the dataset. Values are mean $\\pm$ standard deviation, except for AP, and for FID, which is an aggregate distributional score and not applicable to the mean image baseline.
> AP (IoU=0.50:0.95, all, maxDets=100) measured on stage-wise reconstructions used as inputs to DETR.
> CLIPScore was evaluated with CLIP ResNet-101 and CLIP ViT-L/14 encoders; the table reports ResNet-101 results, and ViT-L/14 exhibited similar trends.
>
> | **Model** | **MSE** $(\\downarrow)$ | **SSIM** $(\\uparrow)$ | **PSNR** $(\\uparrow)$ | **LPIPS** $(\\downarrow)$ | **FID** $(\\downarrow)$ | **CLIPScore** $(\\uparrow)$ | **AP** $(\\uparrow)$ |
> |---|---:|---:|---:|---:|---:|---:|---:|
> | **ViT Modular** ||||||||
> | Backbone | $0.00021 \\pm 0.0001$ | $0.968 \\pm 0.01$ | $37.515 \\pm 2.5$ | $0.036 \\pm 0.02$ | $2.8$ | $0.973 \\pm 0.01$ | $\\times$ |
> | Encoder | $0.008 \\pm 0.003$ | $0.523 \\pm 0.1$ | $21.213 \\pm 1.9$ | $0.493 \\pm 0.045$ | $57.9$ | $0.805 \\pm 0.1$ | $\\times$ |
> | **DETR Modular** ||||||||
> | Backbone | $0.019 \\pm 0.009$ | $0.498 \\pm 0.2$ | $17.820 \\pm 2.2$ | $0.577 \\pm 0.1$ | $110.6$ | $0.756 \\pm 0.1$ | $0.045$ |
> | Encoder | $0.046 \\pm 0.02$ | $0.447 \\pm 0.2$ | $13.844 \\pm 2.0$ | $0.626 \\pm 0.1$ | $114.2$ | $0.746 \\pm 0.1$ | $0.045$ |
> | Decoder | $0.094 \\pm 0.1$ | $0.374 \\pm 0.1$ | $10.905 \\pm 2.3$ | $0.703 \\pm 0.1$ | $156.1$ | $0.706 \\pm 0.1$ | $0.025$ |
> | Pred$_{\\text{FD}}$ | $0.101 \\pm 0.1$ | $0.360 \\pm 0.1$ | $10.570 \\pm 2.3$ | $0.719 \\pm 0.1$ | $165.3$ | $0.699 \\pm 0.1$ | $0.013$ |
> | **DETR Classic (End-to-End)** ||||||||
> | Backbone | $0.018 \\pm 0.009$ | $0.5 \\pm 0.15$ | $17.862 \\pm 2.22$ | $0.566 \\pm 0.07$ | $107.4$ | $0.77 \\pm 0.05$ | $0.045$ |
> | Encoder | $0.043 \\pm 0.02$ | $0.447 \\pm 0.15$ | $14.132 \\pm 2.00$ | $0.632 \\pm 0.08$ | $134.5$ | $0.721 \\pm 0.05$ | $0.023$ |
> | Decoder | $0.064 \\pm 0.02$ | $0.412 \\pm 0.15$ | $12.328 \\pm 1.92$ | $0.646 \\pm 0.07$ | $309.5$ | $0.67 \\pm 0.04$ | $0.000$ |
> | Pred$_{\\text{FD}}$ | $0.069 \\pm 0.02$ | $0.409 \\pm 0.15$ | $11.967 \\pm 1.87$ | $0.658 \\pm 0.07$ | $345.6$ | $0.664 \\pm 0.48$ | $0.000$ |
> | **Mean image baseline** | $0.077 \\pm 0.03$ | $0.402 \\pm 0.1$ | $11.512 \\pm 1.8$ | $0.659 \\pm 0.1$ | $\\times$ | $0.651 \\pm 0.04$ | $0.000$ |

---

> ### Author Response · Authors · 2025-10-28
> **Update on requested experiments**
>
> As requested, we have conducted additional experiments addressing the raised points (for ease of review, these preliminary (unpolished) results are included in the Appendix of the revised manuscript):
>
> **1. Disentangling architecture from objective in DETR–ViT comparisons**
> As promised, we trained a classifier variant of DETR on ImageNet-1K instead of COCO to examine whether DETR’s prototypical abstraction persists without the detection objective. For details, please refer to Section A in the appendix.
>
> **2. Generality across TVMs**
> To assess the generality of our modular feature inversion method, we applied it to DeiT-III [1] and SWIN [2], analyzing whether ViT-like detail preservation extends beyond original ViT. Also, the local image details observed in ViT provide further insight into why SWIN achieves strong performance with local self-attention alone. For details, please refer to Section B in the appendix.
>
> **References**
>
> [1] Touvron, Hugo, Matthieu Cord, and Hervé Jégou. "Deit iii: Revenge of the vit." European conference on computer vision. Cham: Springer Nature Switzerland, 2022.
>
> [2] Liu, Ze, et al. "Swin transformer: Hierarchical vision transformer using shifted windows." Proceedings of the IEEE/CVF international conference on computer vision. 2021.

---

### Decision · Action_Editor_UcB7 · 2025-11-05

**Recommendation:** Reject

**Additional Comments:**

Taking all factors into account, the action editor does not recommend its acceptance as the claims are not fully supported by accurate, convincing, and clear evidence.

**Audience:**

Yes

**Audience Explanation:**

In its current form, some members of TMLR's audience would be interested in this work.

**Claims And Evidence:**

No

**Claims Explanation:**

As the reviewers pointed out, the claimed contributions on revisiting feature inversion and introducing a novel, modular variation that enables a significantly more efficient application of the technique are not well substantiated. Reviewer RzjC commented that the claimed contributions are primarily descriptive rather than explanatory. Reviewer  FtAT commented that this work primarily focuses on interpretability rather than offering technically or theoretically insightful contributions that could inspire new model architectures or design principles. The proposed inverse model produces visually appealing reconstructions, but the results are largely qualitative and do not lead to concrete training or design implications. Reviewer bxCv, on the other hand, finds this paper interesting.